# Dissection of specific binding of HIV-1 Gag to the 'packaging signal' in viral RNA

**Mauricio Comas-Garcia[1], Siddhartha AK Datta[1], Laura Baker[1], Rajat Varma[2], Prabhakar R Gudla[3†], Alan Rein[1]***

[1]HIV Dynamics and Replication Program, Center for Cancer Research, National Cancer Institute, Frederick, United States; [2]Xencor Inc., Monrovia, United States; [3]Optical Microscopy and Analysis Laboratory, Cancer Research Technology Program, Leidos Biomedical Research, Inc., Frederick, United States

**Abstract** Selective packaging of HIV-1 genomic RNA (gRNA) requires the presence of a *cis*-acting RNA element called the 'packaging signal' ($\Psi$). However, the mechanism by which $\Psi$ promotes selective packaging of the gRNA is not well understood. We used fluorescence correlation spectroscopy and quenching data to monitor the binding of recombinant HIV-1 Gag protein to Cy5-tagged 190-base RNAs. At physiological ionic strength, Gag binds with very similar, nanomolar affinities to both $\Psi$-containing and control RNAs. We challenged these interactions by adding excess competing tRNA; introducing mutations in Gag; or raising the ionic strength. These modifications all revealed high specificity for $\Psi$. This specificity is evidently obscured in physiological salt by non-specific, predominantly electrostatic interactions. This nonspecific activity was attenuated by mutations in the MA, CA, and NC domains, including CA mutations disrupting Gag-Gag interaction. We propose that gRNA is selectively packaged because binding to $\Psi$ nucleates virion assembly with particular efficiency.

*For correspondence: reina@mail.nih.gov

Present address: †Laboratory of Receptor Biology and Gene Expression, Center for Cancer Research, National Cancer Institute, Bethesda, United States

Competing interests: The authors declare that no competing interests exist.

## Introduction

Immature virions of HIV-1 and other retroviruses are assembled from the viral Gag polyprotein. This precursor protein consists of matrix (MA), capsid (CA), nucleocapsid (NC) and p6 domains, as well as two spacer peptides (SP1 and SP2) flanking the NC domain. The MA domain binds Gag to the inner leaflet of the plasma membrane (*Zhou et al., 1994*; *Chukkapalli et al., 2010*; *Barros et al., 2016*), although in the cytoplasm it is bound to RNA (*Kutluay et al., 2014*); the CA domain (together with SP1) drives virion assembly (*Ganser-Pornillos et al., 2007*; *Worthylake et al., 1999*; *Datta et al., 2011a*; *Datta et al., 2015*); the NC domain is required for incorporation of nucleic acids (the two zinc fingers in this domain are necessary for selective RNA packaging) (*Aldovini and Young, 1990*; *Gorelick et al., 1990*; *Cimarelli et al., 2000*; *Zhang et al., 1998*); and the p6 domain interacts with the host-cell machinery to release virions from the cell (*Freed, 2002*; *Usami et al., 2009*). As far as is known, all retrovirus particles also contain RNA. In fact, addition of RNA to recombinant Gag protein in vitro leads to its assembly into virus-like particles (VLPs) (*Campbell and Rein, 1999*; *Campbell et al., 2001*; *Gross et al., 2000*). We have suggested that cooperative binding to nucleic acid brings Gag molecules into close proximity with each other, thereby nucleating virion assembly (*Rein et al., 2011*; *Datta et al., 2011a*; *Datta et al., 2015*; *Comas-Garcia et al., 2016*). In particular, we have found that SP1 undergoes a conformational change when two or more copies are closely juxtaposed (*Datta et al., 2015*). It is plausible that this change causes the exposure of interfaces in Gag required for particle assembly. However, it is not known if the identity of the bound nucleic acid influences this process.

Almost any single-stranded nucleic acid can support virus-like particle (VLP) assembly in vitro (*Campbell and Rein, 1999*). Nonetheless in infected cells, the full-length viral RNA (genomic RNA or gRNA) is encapsidated with very high selectivity. However, when Gag is expressed in mammalian cells lacking gRNA, it assembles into VLPs containing a nearly random assortment of cellular mRNA molecules (*Rulli et al., 2007*). Thus, the 'structural' role of RNA in virion assembly can be provided by either gRNA or by cellular RNAs. In infected cells, the gRNA is competing with a large excess of cellular RNAs and spliced viral RNAs for encapsidation, and evidently has a strong advantage in this competition.

What is the nature of this advantage? As in all retroviruses, the HIV-1 gRNA contains a *cis*-acting RNA element called the 'packaging signal' ('Ψ') that is required for its selective packaging during virion assembly (*Aldovini and Young, 1990*; *Lever et al., 1989*; *Clavel and Orenstein, 1990*). This signal consists of several hundred nucleotides in the 5'-untranslated region (UTR) and part of the Gag open reading frame. However, how the presence of Ψ leads to preferential encapsidation is not understood. One possibility is that Ψ is a high-affinity binding site for Gag. In fact, a great deal of effort has been devoted to measuring binding affinities of Gag and NC for nucleic acids (*Shubsda et al., 2002*; *Webb et al., 2013*; *Fisher et al., 2006*; *Abd El-Wahab et al., 2014*; *Bernacchi et al., 2017*; *Cruceanu et al., 2006*; *Dannull et al., 1994*; *Berkowitz et al., 1993*). However, we show here that under physiologically relevant salt concentrations, this protein binds to Ψ and non-Ψ RNAs with very similar, nanomolar affinities. Thus, it seems unlikely that a simple high-affinity scenario could explain the packaging selectivity observed in vivo.

In the present work, we have investigated the question of whether Gag discriminates between Ψ and non-Ψ RNAs at the level of binding. As noted above, we find that at near-physiological ionic strengths the affinities of Gag for these different RNAs are virtually indistinguishable. However, when the strength of Gag-Gag and/or Gag-RNA interactions is modulated, Gag binds with high specificity to a Ψ-containing RNA. While there are both specific and non-specific components in binding to any RNA, there is a much larger contribution of specific binding with Ψ. Unexpectedly, strong Gag-Gag interactions make a major contribution to the non-specific component. We suggest a novel general mechanism by which the type and strength of Gag-Gag and Gag-RNA interactions during binding might lead to preferential encapsidation of gRNA.

## Results

The goal of the experiments described here was to measure and characterize the binding of recombinant Δp6 HIV-1 Gag protein to short single-stranded RNAs. As noted in the Introduction, addition of RNA to Gag can lead to assembly of VLPs, complicating the measurement of a dissociation constant ($K_D$) for the binding. In fact, most biophysical techniques (e.g., sedimentation experiments or isothermal calorimetry) require high Gag concentrations (micromolar regime); at these concentrations Gag/RNA complexes will associate into virus-like-particles. We have circumvented this problem by using FCS, in which the concentration of the fluorophore-tagged RNA is low enough that association of Gag/RNA complexes into VLPs or other large structures is not thermodynamically favorable.

We have analyzed the binding of Gag to three 175-base viral-derived RNAs: HIV-1 nt 193–368, referred to here as 'Ψ'; HIV-1 nt 2004–2179, called 'GRPE'; and Moloney MLV nt 202–377, called 'MoMLV Ψ'. GRPE has been suggested to contribute to selective packaging of gRNA (*Chamanian et al., 2013*), but this now seems unlikely (*Nikolaitchik et al., 2014*). As many lines of evidence indicate that only dimeric gRNA is selectively packaged (*Moore et al., 2009*), we studied the HIV-1 Ψ RNA in both monomeric and dimeric forms (see Materials and methods). A 15-nt long polyA tail was added at the 3'-end of the viral sequences to increase the labeling efficiency. The RNAs were covalently labeled at their 3' ends with Cy5, as detailed in Materials and methods.

In a typical experiment, the RNA was diluted in Binding Buffer to a concentration of 15 nM and different amounts of Gag protein (final concentrations between 15 and 300 nM) were added. After overnight incubation at 4° C, the mixtures were analyzed by FCS at room temperature. The autocorrelation curves from Cy5-labeled RNAs were analyzed to obtain the weighted average diffusion constant ($D$) of the RNAs in each mixture. In addition, we found that binding of Gag induced quenching of the Cy5 fluorophore; thus the fraction of RNA molecules bound to Gag could be estimated from the normalized fluorescence intensity in each reaction mixture. It has been shown that quenching of this fluorophore results from the steric stabilization of a non-fluorescent photon-induced Cy5 isomer

(*Levitus and Ranjit, 2011*; *Stennett et al., 2014*). Thus, by using FCS, we can determine, in a single experiment, changes in coarse-grained structure (i.e., overall size) of the RNA and the fraction of bound and free RNAs. First we will discuss the effects of Gag binding upon the overall structure of Gag/RNA complexes and then the contributions of the different domains of the polyprotein Gag to RNA binding affinity and specificity.

## Gag binding causes RNA collapse

Binding of Gag to the tested RNAs resulted in an increase in the $D$ of the RNAs. In *Figure 1a* it can be seen that the $D$ for each RNA increases as the concentration of Gag is increased, reaching a plateau at $\approx$ 50–100 nM Gag in the case of HIV-1 $\Psi$ (monomeric and dimeric) and $\approx$ 200–250 nM Gag with MoMLV $\Psi$ and GRPE RNAs. (*Figure 1—figure supplement 4* shows that the diffusion coefficients ($D$) of the monomeric and dimeric HIV-1 $\Psi$ RNAs were different enough to be resolved from each other by FCS and that the dimerization efficiency was $\approx$ 90%). Thus, the binding of Gag results in a decrease in the hydrodynamic radius of the RNA. This was somewhat unexpected, as the Gag/

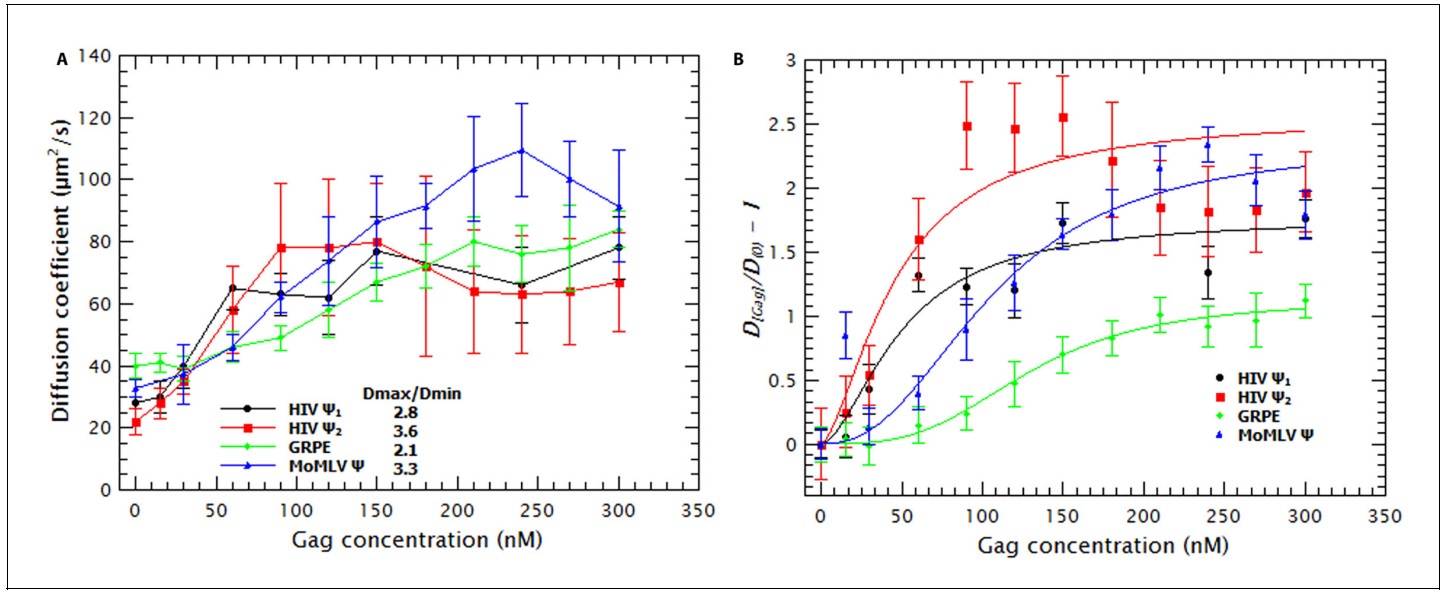

**Figure 1.** Binding of Gag collapses the RNA. (**A**) The diffusion coefficient ($D$) of the Cy5-labled RNAs increases with increasing Gag concentration. The ratio of the maximum diffusion coefficient of bound ($D_{max}$) and free RNA ($D_{min}$) quantifies the degree of RNA collapse. $\Psi_1$, monomeric $\Psi$; $\Psi_2$, dimeric $\Psi$. (**B**) The normalized ratio of the maximum diffusion coefficient of bound ($D_{[Gag]}$) and free RNA ($D_{min}$) as a function of Gag concentration is best fit with a simple cooperative model (lines). The quality of the RNAs was tested by running denaturing gels (see *Figure 1—figure supplement 1a*) while native gels were used to verify that the dimerization protocol did not produce large aggregates (see *Figure 1—figure supplement 1b*). Sedimentation velocity and FCS were used to verify that there were no detectable Gag (*Figure 1—figure supplement 2*) or RNA (*Figure 1—figure supplement 3*) aggregates due to freezing/thawing cycles. A comparison of the distribution of the diffusion coefficients of the HIV $\Psi$ that had been thermally annealed to produce either monomers or dimers is shown in *Figure 1—figure supplement 4*. *Figure 1—figure supplement 5* shows that addition of Proteinase K to Gag/RNA complexes decreases the $D$ to that of free RNA.

The following figure supplements are available for figure 1:

**Figure supplement 1.** The quality of the RNAs was determined by gel electrophoresis.

**Figure supplement 2.** Freezing/thawing cycles of Gag does not yield protein aggregation.

**Figure supplement 3.** Freezing/thawing cycles and thermal annealing of the RNAs do not result in RNA aggregation.

**Figure supplement 4.** The thermally annealed HIV-1 $\Psi$ dimers can be resolved from monomers by FCS and are stable at 15 nM.

**Figure supplement 5.** Proteolytic digestion of Gag/RNA complexes recovers the diffusion coefficient of the RNA.

RNA complexes must have greater masses than the free RNAs, and implies that binding of Gag to the RNA causes a collapse in the RNA. The increase in the diffusion coefficient is not due to RNA degradation; upon digestion of the Gag/RNA complexes with Proteinase K, the **D** of the labeled RNA reverts to that of naked RNA (see *Figure 1—figure supplement 5*). Capsid proteins of other RNA viruses also cause RNA collapse (*Borodavka et al., 2012*, *2013*).

Molecular dynamics simulations indicate that RNA collapse by capsid proteins involves protein-protein interactions (*Perlmutter et al., 2014*). If RNA collapse by capsid proteins is driven by multiple protein molecules interacting with each other as well as with the RNA, then it should be a cooperative process. To test this hypothesis, we normalized the diffusion constants of the RNAs in the titrations relative to that of pure RNA, so that the normalized **D** of pure RNA was set to zero (*Figure 1B*). These plots are best fitted by using a model in which the change in **D** as a function of Gag concentration is cooperative: the exponent associated with the Gag concentration required for RNA collapse (analogous to the Hill coefficient ($n_H$) in the Hill Equation for cooperative binding) was greater than one for all RNAs (between 1.7 and 3.4). Therefore, Gag-induced RNA collapse is evidently a cooperative process. It should be pointed out that the fit for the dimeric HIV-1 $\Psi$ is relatively poor, probably because the number of data points in the sigmoidal portion of the curve is very small.

## Gag binds to RNAs with high affinity but low specificity

As mentioned before, the fluorescence of the RNAs is reduced by addition of Gag (see *Figure 2A*). By using the approach described in the supplementary information, the normalized fluorescence data in *Figure 2A* were used to estimate the fraction of RNA molecules bound by Gag as well as the binding mechanism. The resulting binding curves are presented in *Figure 2B*. It is obvious that, at 0.2 M NaCl, the affinities of Gag for the different RNAs are all very similar to each other. That is, the concentrations of Gag needed to bind half of the RNAs in solution for the different RNAs are all within a factor of two of each other.

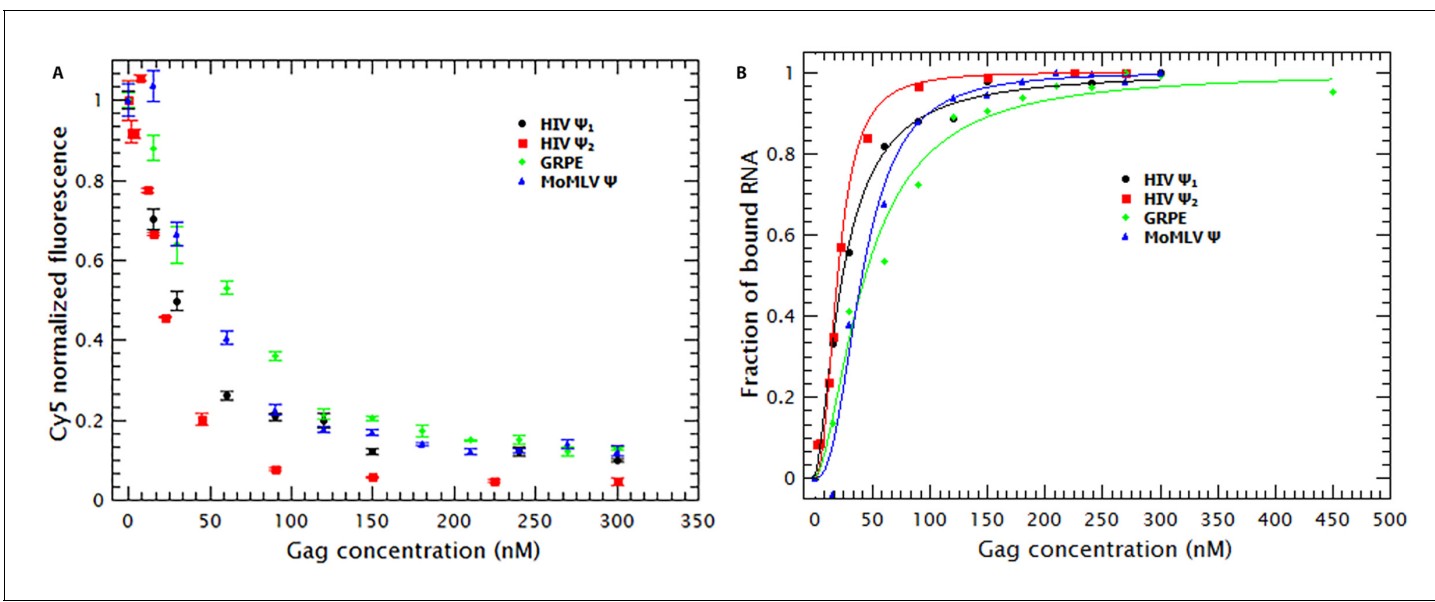

**Figure 2.** Binding of Gag quenches Cy5; quenching is a metric of binding. (**A**) The fluorescence intensity of Cy5 decreases with increasing Gag concentration. (**B**) Binding plots of wt Gag with different RNAs, derived from quenching data in A. The binding plots are best fit to a cooperative model (lines). The binding plots show that the dimeric HIV-1 $\Psi$ (HIV $\Psi_2$) has the highest binding affinity. However, the difference in the binding affinity between this RNA and the other RNAs is minimal. *Figure 2—figure supplement 1* shows a representative comparison of three different binding models; cooperative, multiple non-cooperative and non-cooperative binding.

The following figure supplement is available for figure 2:

**Figure supplement 1.** Binding of Gag to RNA is cooperative.

To obtain a binding mechanism and quantitate the affinities, the binding plots were fitted to several binding models: cooperative binding (Hill equation), non-cooperative multiple binding and one-to-one binding (see supplementary information). *Figure 2—figure supplement 1* is a representative example that shows that the data is best described when fitted with a simple cooperative model (Hill equation). By fitting the binding curves in *Figure 2B* to this model we estimated the $K_D$ and $n_H$ for each RNA. *Table 1* shows that the $K_D$s obtained ranged from 21 nM for dimeric HIV-1 Ψ to 44 nM for GRPE. Furthermore, the $n_H$s derived from these curves varied between 1.6 and 2.5 (see *Table 1*), implying that cooperativity is not a specific property of the RNA but a general characteristic of Gag. (These measurements are probably not precise enough to distinguish between the levels of cooperativity obtained with different proteins and/or different RNAs.) This binding model is consistent with the cooperativity of the Gag-induced RNA collapse. Finally, we see that under these experimental conditions Gag binds to RNAs with high affinity but low specificity. The difference in the binding affinity between the dimeric HIV-1 Ψ and the control RNAs appears insufficient to explain selective RNA packaging in vivo.

## The presence of a competitor RNA reveals binding specificity for the dimeric HIV-1 Ψ

To test the in vitro binding specificity of Gag for the HIV-1 Ψ we repeated the titrations in the presence of an excess of an irrelevant competitor RNA, *viz.* yeast tRNA. Unless otherwise noted, in the competition experiments the tRNAs and Gag were pre-incubated together for about 10–15 min before addition of the labeled RNAs. The concentration of the tRNAs was 50-fold higher by mass than that of the labeled RNAs (≈120-fold molar excess).

As shown in *Figure 3*, the binding curves obtained in the presence of tRNAs showed that addition of these competitor RNAs causes only a modest reduction in the binding to the monomeric and dimeric HIV-1 Ψ RNAs, a somewhat larger reduction in binding to MoMLV Ψ, and a drastic decrease in the binding to GRPE RNA. The $K_D$s derived from these curves (*Table 1*) show that the apparent affinities of Gag for both forms of HIV-1 Ψ were reduced roughly ≈2-fold by the addition of the tRNAs, while those for the MoMLV Ψ and GRPE were reduced ≈4.5 and 7-fold, respectively. These data show that Gag has a higher binding specificity for the HIV-1 Ψ than for the control RNAs.

## tRNAs reduce the degree of Gag-induced RNA collapse

We also found that the presence of tRNAs partially suppressed the Gag-induced increase in the diffusion constants of the tagged RNAs (see *Figure 3—figure supplement 1A*). In fact, the diffusion plots for Gag titrations of the MoMLV Ψ and the GRPE RNAs (broken blue and green data in *Figure 3—figure supplement 1A*, respectively) show that the presence of tRNAs resulted in a *D* similar to that of the naked RNAs. This decrease in RNA collapse is not due to interference with binding of

**Table 1.** Binding parameters for Gag/RNA interactions.
$K_D$ and $n_H$ values were obtained by fitting the binding curves as described in Materials and methods and in the Supplementary Information.

| Sample | tRNA | WT-Gag | | 8N-Gag | | WM-Gag | |
|---|---|---|---|---|---|---|---|
| | | $K_D$ (nM) | $n_H$ | $K_D$ (nM) | $n_H$ | $K_D$ (nM) | $n_H$ |
| HIV Ψ₁ (monomeric) | - | 21.4 | 3.2 | 60 | 2.4 | 31.9 | 2.4 |
| HIV Ψ₁ (monomeric) | + | 53.3 | 1.7 | 60 | 2.4 | 124.8 | 3.7 |
| HIV Ψ₂ (dimeric) | - | 20.3 | 2.4 | 43.2 | 2.1 | 15.2 | 2.4 |
| HIV Ψ₂ (dimeric) | + | 37 | 2.1 | 54.7 | 2.8 | 55.8 | 2.2 |
| MoMLV Ψ | - | 25.9 | 2.1 | 59.8 | 2.5 | 59 | 4.6 |
| MoMLV Ψ | + | 118.8 | 3.5 | 59 | 4 | ≥300 | - |
| GRPE | - | 44.2 | 1.7 | 145 | 3.9 | 111.8 | 3.8 |
| GRPE | + | 315.4 | 2.8 | 154.3 | 3.8 | >>300 | - |

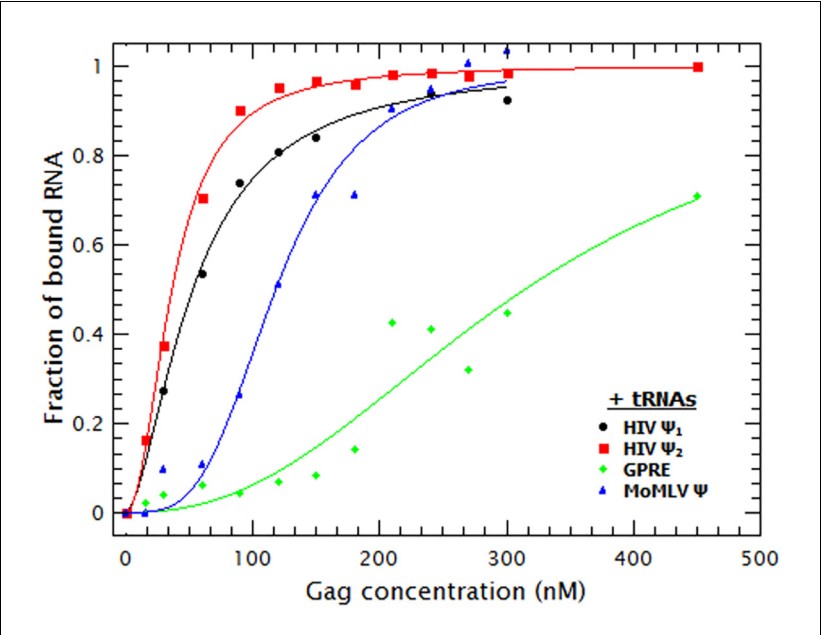

**Figure 3.** Addition of a competitor RNA reveals binding specificity. Addition of a large excess of yeast tRNAs ($\approx$120 moles of tRNA per mole of RNA) greatly increases the difference in the apparent affinity between that for the dimeric HIV-1 $\Psi$ ($\Psi_2$) (red sequares) and that for the other RNAs. The data was fitted with a cooperative binding model (solid lines). *Figure 3—figure supplement 1* shows how both the diffusion coefficient of the RNAs and the Cy5 quenching data are affected by the presence of tRNAs.
The following figure supplement is available for figure 3:

**Figure supplement 1.** The presence of tRNAs increases the size of the Gag/RNA complexes.

Gag to the RNAs: with the exception of GRPE RNA the quenching plots in the competition experiments exhibited a plateau, just as in the absence of tRNAs (see *Figure 3—figure supplement 1B*). The presence of a plateau means that further addition of Gag does not result in more binding and implies that all of the RNAs in solution are bound. Moreover, at least for the dimeric HIV-1 $\Psi$ at Gag concentrations above 100 nM, the degrees of quenching in the presence and absence of tRNAs are indistinguishable from each other (see *Figure 3—figure supplement 1B*) It seems possible that tRNAs attenuate the Gag-induced increase in $D$ because they are also incorporated into the Gag/RNA complexes.

## Binding hysteresis

Cooperativity of binding (*Figure 2* and *Table 1*) implies that Gag molecules can interact with each other while bound to an RNA. This adds a degree of complexity to the binding equilibria. In the competition experiments first Gag and the tRNAs were mixed and then the tagged RNAs were added; therefore, addition of the tagged RNAs may lead to a re-equilibration process. It is possible that the presence of protein-protein interactions could introduce binding hysteresis so that the apparent $K_D$ for the competition experiment could depend on the order of addition of the RNAs.

To test this possibility, we did competition experiments with the monomeric HIV $\Psi$ and GRPE in which the tRNAs were added either before or after mixing Gag with the labeled RNAs. As a control we titrated Gag into these RNAs in the absence of tRNAs. The binding plots for these experiments are shown in *Figure 4A*. On the one hand, when the tRNAs were added after mixing Gag and monomeric HIV-1 $\Psi$ (red diamonds), the apparent affinity for $\Psi$ was very similar to that in the absence of tRNAs (red circles) ($K_D$s of 29 vs 26 nM, respectively), while the inverse order of addition showed only a slight decrease in the apparent affinity ($K_D$ of 44 nM) (red squares). On the other hand, the apparent affinity for the GRPE (green diamonds and squares) is greatly affected by order of addition

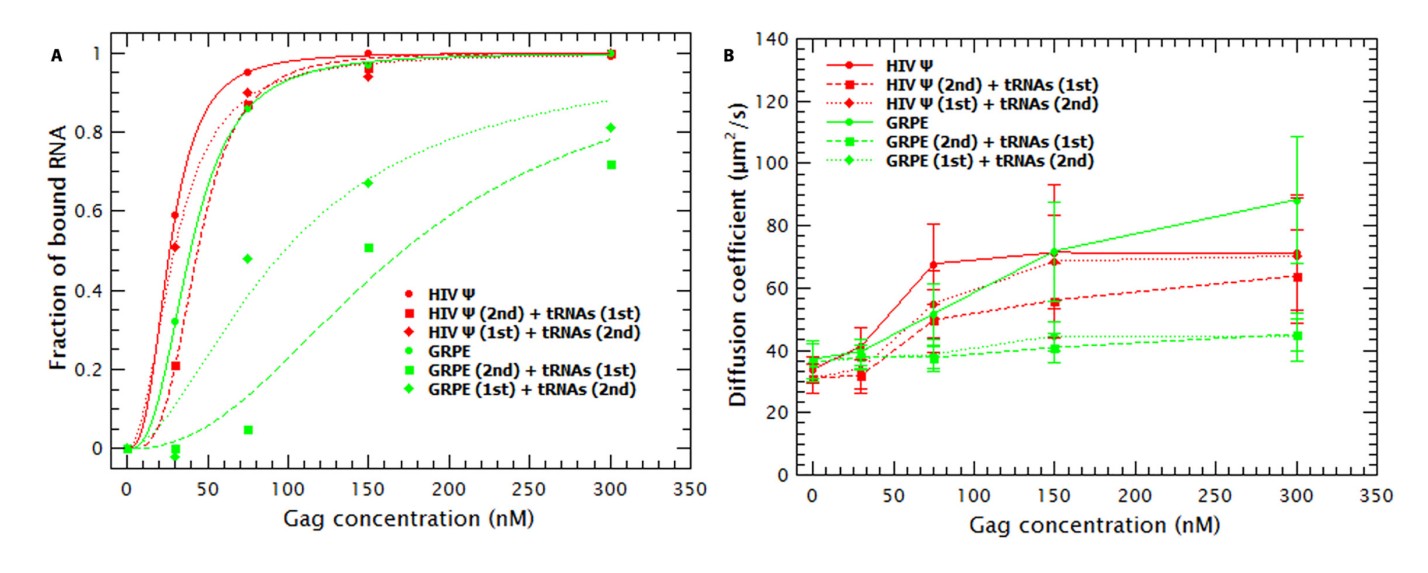

**Figure 4.** Hysteresis in Gag/RNA binding. (A) The presence of excess tRNAs and the order in which the tRNAs are added to the reaction mixture with the monomeric HIV-1 Ψ have a minor impact in the binding plots (red data). In contrast, with GRPE RNA the order of addition, as well as the presence of excess tRNAs, has a large influence on the binding isotherm. (B) Addition of tRNAs after mixing HIV-1 Ψ and Gag (red diamonds) only blocks RNA collapse at low Gag concentrations. tRNAs block GRPE collapse independent of the order in which they are added (green squares and diamonds).

of the RNAs. Unlike the results with HIV-1 Ψ, addition of tRNAs reduced the apparent affinity of Gag for GRPE by 2.8-fold when added after the GRPE, but 5.5-fold if added prior to GRPE. (It should be noted that in these experiments there are fewer data points than in those of *Figure 3*, and hence there is a larger error in the $K_D$s for competition experiments with the GRPE RNA in *Figure 4A* than in *Figure 3*.) These results show that the binding hysteresis is greater for the GRPE than for the HIV-1 Ψ.

As mentioned before, incubation of tRNAs with Gag prior to addition of labeled RNAs increased the size of the Gag/RNA complexes (see *Figure 3—figure supplement 1A* and red data in *Figure 4B*). The effect of the tRNAs on the size of the Gag/HIV-1 Ψ complexes also depends upon the order of addition of the two species. As shown in *Figure 4B*, addition of tRNAs to pre-incubated Gag/HIV-1 Ψ complexes only affects the size of these complexes at low Gag concentrations. However, independent of the order of addition, the presence of tRNAs blocks the Gag-induced collapse of the GRPE RNA. The difference in the degree of hysteresis in both the binding plots (*Figure 4A*) and the diffusion plots (*Figure 4B*) implies that addition of tRNAs has a weaker effect on Gag/HIV-1 Ψ complexes than on the Gag/GRPE complexes. This suggests that Gag/HIV-1 Ψ complexes are more stable than Gag/GRPE complexes.

## MA-RNA interactions contribute to non-specific binding

Of all the domains of Gag the NC is perhaps the most studied in terms of its interactions with nucleic acids (*Cimarelli et al., 2000*; *Shubsda et al., 2002*; *Fisher et al., 2006*; *Cruceanu et al., 2006*; *Dannull et al., 1994*; *Berkowitz et al., 1993*). As disruption of the zinc fingers in this domain abrogates RNA packaging selectivity (*Aldovini and Young, 1990*; *Gorelick et al., 1990*), these motifs are crucial for gRNA binding and packaging during virion assembly. However, the roles of the MA and CA domains in RNA binding specificity are much less understood.

First, we assessed the contributions of the MA domain to RNA-binding and specificity by assaying binding of '8N Gag'. In this mutant Gag protein, eight basic residues in the N-terminal region of the MA domain have been replaced with asparagines (*Zhou et al., 1994*). *Table 1* shows that removing these charges in MA reduced the affinity of Gag for monomeric or dimeric Ψ by 2.8 and 2.2-fold, respectively, and 2.3 and 3.3-fold for the MoMLV Ψ and GRPE RNA, respectively. *Figure 5A* compares the binding plots for wild-type (WT) (solid symbols) and 8N Gag (open symbols). The 8N

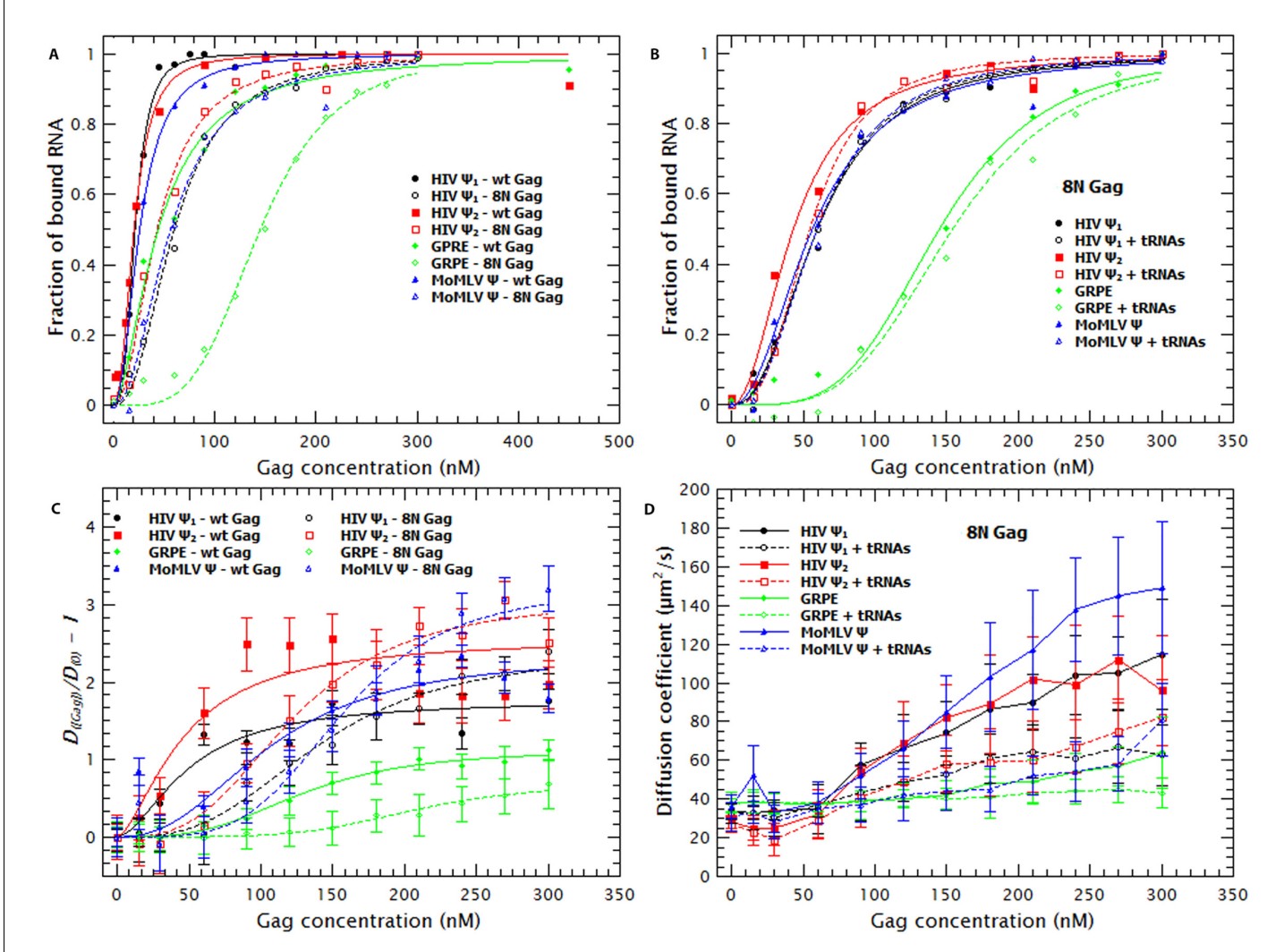

**Figure 5.** The Matrix domain of Gag contributes to non-specific binding but has no major impact on RNA collapse. (**A**) Side-by-side comparison of the binding plots for WT (solid symbols) and 8N Gag (open symbols) to monomeric $\Psi$ (circles), dimeric $\Psi$ (squares), GRPE (diamonds) and MoMLV $\Psi$ (triangles). Reducing the charge in MA affects the binding affinity for these RNAs. (**B**) The comparison of the binding plots for 8N Gag in the absence (solid symbols) and presence (open symbols) of a $\approx$ 120-fold molar excess of tRNAs. (**C**) Comparison of the normalized diffusion plots for the different RNAs upon binding of WT (solid symbols) and 8N Gag (open symbols). The lines are a cooperative fit of the Gag-induced RNA collapse. (**D**) As with WT Gag (solid symbols), the presence of excess tRNAs increases the size of the 8N Gag/RNA complexes (open symbols).

mutation decreased the binding affinity for GRPE RNA, but had little effect on the affinities for the other RNAs. There is evidence that in cytoplasmic Gag, the MA domain is largely bound to tRNAs (*Kutluay et al., 2014*). Thus it was of interest to determine whether the 8N mutation would alter the effects of tRNA addition upon binding to the tagged RNAs. As shown in *Figure 5B* and *Table 1*, we found that the tRNA had no detectable effect on the apparent binding affinity of 8N Gag for the tagged RNAs. Thus, the eight positive residues in MA that are lacking in 8N Gag are essential for the binding to tRNAs, at least when in competition against these viral-derived RNAs.

*Figure 5C* shows that binding of 8N Gag to the RNA also results in an increase in the diffusion coefficient of the RNA. With the exception of the GRPE RNA, the magnitude of the increase in the diffusion coefficient of the RNAs ultimately induced by binding to 8N Gag (open symbols) is greater than that induced by WT Gag (solid symbols). However, 8N Gag requires about 1.5–2.8 fold more protein than the wild-type protein to induce this collapse in the RNAs. As with WT Gag, RNA collapse by 8N Gag also fits a cooperative model (i.e., $n_H$ >1) (broken lines in *Figure 5C*).

Interestingly, while addition of tRNA did not reduce the apparent affinity of 8N Gag for the RNAs (*Figure 5B*), it did increase the size of the resulting complexes (see *Figure 5D*). Evidently, the eight basic residues that are absent in 8N Gag are necessary for competitive binding by tRNA, but they are not required for incorporation of tRNAs into Gag-RNA complexes. It should be noted that the change in *D* of the RNAs, but not in apparent affinity, of 8N Gag induced by tRNA shows that the increase in *D* is not an artifact due to Cy5 quenching.

## Specific binding depends on the strength of Gag-Gag interactions

In solution Gag is in monomer-dimer equilibrium with a $K_D \approx 10$ μM (*Datta et al., 2007b*). The dimeric interaction between the C-terminal domain (CTD) in the CA domain drives this equilibrium and contributes to assembly of both immature and mature virions in vivo. To determine the role, if any, of Gag dimerization and the Gag dimer-interface in RNA-binding, we characterized the binding affinity of a Gag protein with impaired dimerization capabilities. Gag with the double mutation W184A/M185A ('WM Gag') has a dimerization affinity ≈100X lower than WT Gag (*Datta et al., 2007a*, *2007b*). A comparison of the binding plots with WT (closed symbols) and WM Gag (open symbols) is shown in *Figure 6A*. These plots, as well as the $K_D$s in *Table 1*, show that the affinity of WM Gag for the dimeric HIV-1 Ψ is roughly the same as that of the wild-type protein. The affinities of WM and wild-type Gag for the monomeric Ψ and for MoMLV Ψ are also very similar to each other. In contrast, the affinity of this mutant protein for GRPE RNA is ≈2.5-fold lower than that of wild-type Gag. Interestingly, the Hill coefficient for WM Gag/RNA binding was still greater than one.

As shown in *Figure 6—figure supplement 1*, WM Gag induces an increase in the *D* of HIV-1 and MoMLV Ψ RNAs, but these effects were considerably smaller than those caused by WT Gag. There was no detectable change in the *D* of GRPE RNA, despite the evidence from quenching showing that there is still binding to this RNA (see *Figure 6A*). Interestingly, the $n_H$s of the fits for the collapse of the other RNAs are still greater than one (≈1.7). These results confirm that Gag-Gag

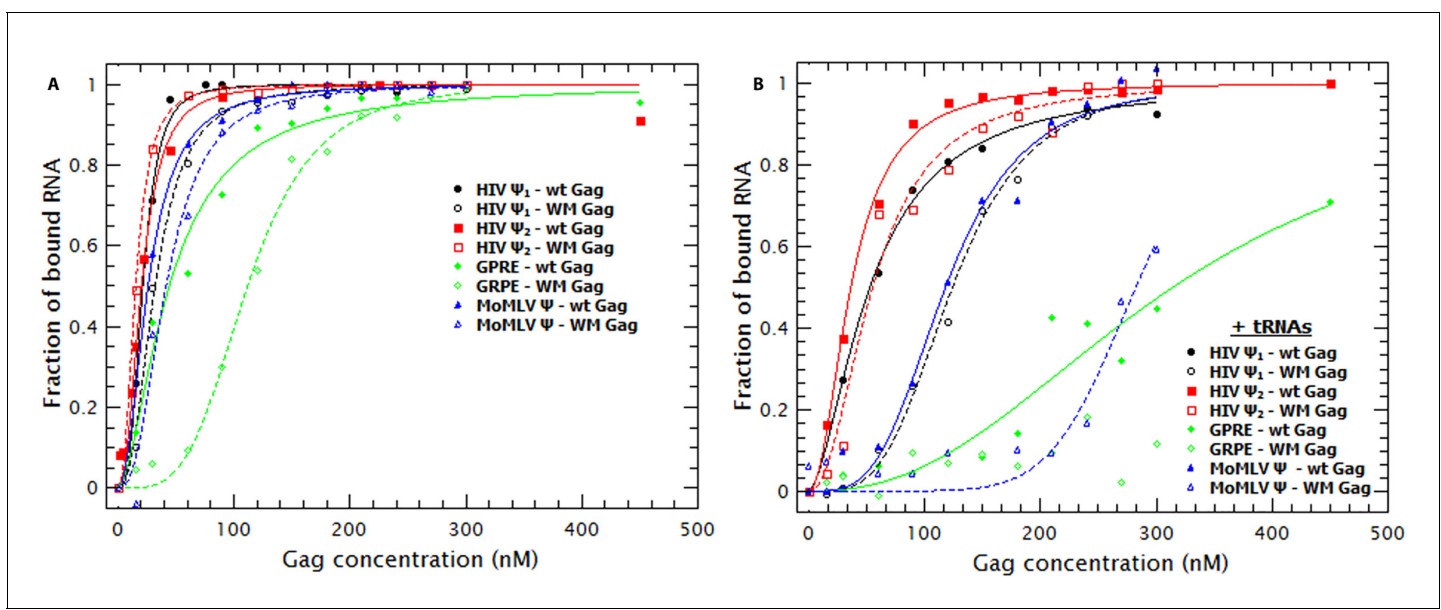

**Figure 6.** Non-specific binding is greatly decreased by decreasing the strength of Gag-Gag interactions. (**A**) Comparison of the binding plots of WT (closed symbols) and WM Gag (open symbols). While the binding affinity for the monomeric and dimeric HIV-1 Ψ (black and red symbols, respectively) as well as the MoMLV Ψ (blue triangles) is not greatly affected by the WM mutation, binding to the GRPE RNA (green diamonds) is significantly impaired. (**B**) In the presence of tRNAs binding to the GRPE (green diamonds) and MoMLV Ψ (blue triangles) is greatly impaired while binding to the dimeric HIV-1 Ψ (red squares) suffers only a minimal decrease in binding affinity. *Figure 6—figure supplement 1* compares the degree of RNA collapse by WT and WM Gag; RNA collapse by the mutant protein is impaired.

The following figure supplement is available for figure 6:

**Figure supplement 1.** Reducing Gag-Gag interactions decreases the degree of RNA collapse.

interaction plays a major role in RNA collapse; taken together with the cooperativity of the binding of WM Gag (*Figure 6A*, *Table 1*), they suggest that this interaction is partially independent of the dimer interface in the CTD.

Incubation of yeast tRNAs with WM Gag prior to the addition of the labeled RNAs (see open symbols in *Figure 6B*) had a moderate impact on the apparent affinity for the HIV-1 Ψ RNAs and a large effect on the apparent affinity for the control RNAs compared to the effect seen with wild-type Gag (filled symbols in *Figure 6B*). The presence of tRNAs increased the apparent $K_D$ of WM Gag for the monomeric and dimeric Ψ from 32 to 125 nM and from 15 to 56 nM, respectively. This effect was more pronounced on the control RNAs; the $K_D$ of WM Gag for the MoMLV RNA increased from 59 to >300 nM and we were not able to detect binding to the GRPE RNA. These results clearly demonstrate that when Gag-Gag interactions are impaired, Gag retains high affinity for dimeric HIV-1 Ψ, but not for the control RNAs.

## Decreasing electrostatic interactions increases binding specificity

Binding of a protein to nucleic acids could be driven by non-specific interactions, such as the electrostatic attraction between positively charged side chains in the protein and the phosphate groups on the RNA. It could also be a highly specific interaction; in most cases, these interactions are non-electrostatic. Finally, it might be a combination of both specific and non-specific interactions (*Record et al., 1978*, *1976*; *Rouzina and Bloomfield, 1997*). The role of electrostatic interactions can be assessed by varying the ionic strength at which the binding is measured: cations in the solution will compete with positive charges in the protein for binding to the negatively charged RNA. The reduction in binding as the ionic strength is increased indicates the relative contribution of electrostatic interactions to binding. Using fluorescence anisotropy to monitor binding, Webb *et al.* determined the relative contribution of these interactions by salt-titrating Gag/RNA complexes (*Webb et al., 2013*). By using this approach Webb and coworkers extrapolated the $K_D$ of Gag for several short RNAs to 1 M NaCl ($K_{D(1M)}$) and estimated the net number of ions displaced during binding. The $K_{D(1M)}$ represents the dissociation constant in the absence of electrostatic interactions, which can be understood as the contribution of specific interactions to the overall $K_D$.

To understand the contribution of electrostatic and non-electrostatic interactions to binding we performed Gag titrations at different NaCl concentrations with the RNAs that exhibited the highest

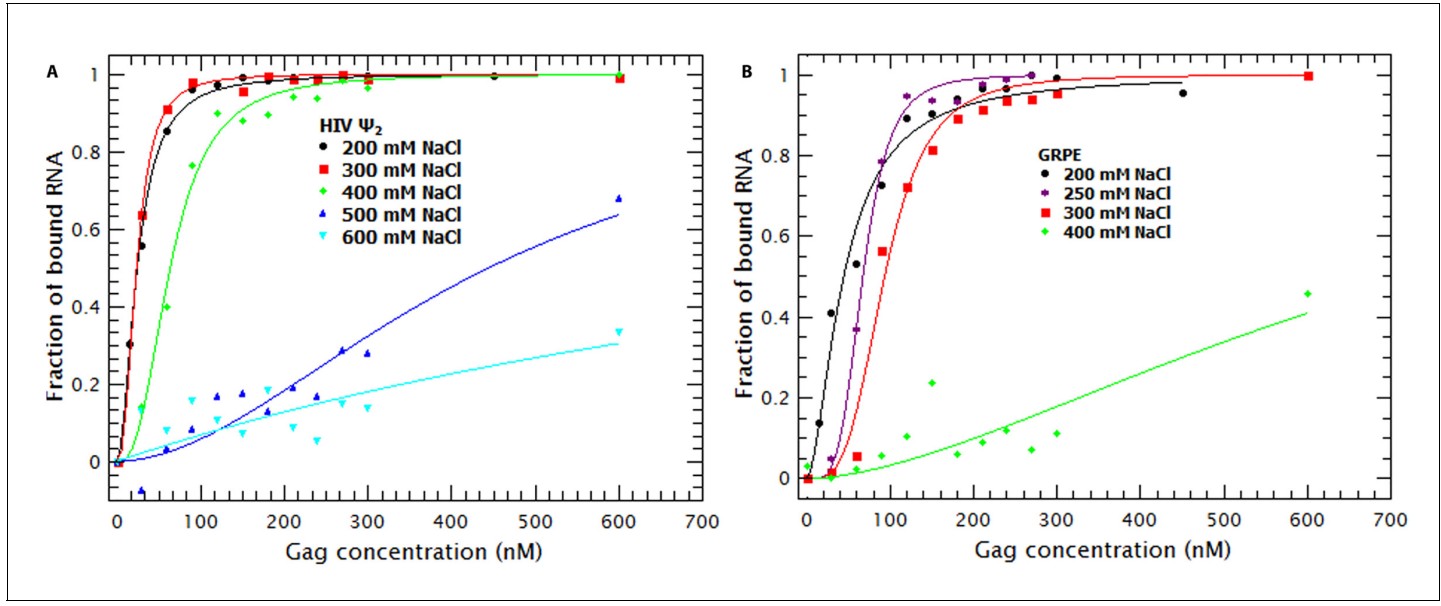

**Figure 7.** Binding of Gag to dimeric HIV-1 Ψ has a larger non-electrostatic component than binding to GRPE RNA. (**A**) Binding of WT Gag and the dimeric HIV-1 Ψ (Ψ₂) was measured at various ionic strengths. Increasing the salt concentration from 0.2 to 0.4 M NaCl had a minor effect on the binding affinity of the dimeric Ψ. Binding to dimeric HIV-1 Ψ (Ψ₂) was detectable as high as 0.6 M NaCl. (**B**) The highest salt concentration at which we could measure binding to the GRPE RNA was 0.4 M NaCl.

and lowest affinity for Gag: the dimeric HIV Ψ and GRPE, respectively. *Figure 7A and B* shows the binding curves for the dimeric HIV Ψ and the GRPE at various ionic strengths. It is obvious that the interactions of Gag with these two RNAs are profoundly different from each other with respect to their dependence on the ionic strength. Increasing the NaCl concentration from 200 to 300 mM had no detectable effect on the dimeric HIV-1 Ψ, while it did decrease the affinity of Gag of the GRPE RNA. Furthermore, at 400 mM NaCl there is only a small decrease in the affinity of Gag for Ψ, but a dramatic loss in binding to GRPE. Binding to Ψ was still detectable, although reduced, when the NaCl concentration was raised to 500 or 600 mM.

To quantify the relative contribution of electrostatic and non-electrostatic interactions to Gag/RNA binding, we plotted the $K_D$s calculated from *Figure 7* as a function of salt concentration in a log-log plot (see *Figure 8*). According to biophysical studies in other systems (*Record et al., 1976*), the slope of this plot represents the net number of ions displaced by binding and the *y*-intercept represents $K_{D(1M)}$. Surprisingly, with our data, these plots are not linear (see *Figure 8*); rather, they each have a rather shallow slope at low salt concentration (1.6 and 1.8 for the dimeric Ψ and GRPE, respectively) and a steeper slope at higher ionic strengths (7.6 and 7.2, respectively). The experimental variation in our binding assays is ≈ 10% and the difference between the last data point in the shallow slope (i.e., at 0.4 M NaCl for the dimeric HIV-1 Ψ) and first point in the steep slope (i.e., 0.5 M NaCl for the dimeric HIV-1 Ψ) is 7-fold. Hence, the appearance of a second slope is not an artifact due to the uncertainty of the measurement. The non-linearity of these plots suggests that Gag has two different binding modes, involving different numbers of ions, at different ionic strengths. Possible explanations for the non-linearity are discussed below.

## Decreasing NC-RNA or Gag-Gag interactions reveals specific binding at slightly higher than physiological salt concentrations

To further dissect the contribution of protein-protein and protein-RNA interactions to specific binding we tested the effects of increasing ionic strength upon binding to different mutant Gag proteins. Binding curves and diffusion plots for both dimeric Ψ and GRPE RNAs were obtained for each protein at different NaCl concentrations (see *Figure 9—figure supplement 1* and *Figure 9—figure supplement 2*). The $K_D$s for each protein-RNA combination over a range of salt concentrations for the dimeric HIV-1 Ψ and GRPE RNA are shown in *Figure 9A and B*, respectively. As detailed below, we also analyzed the interaction of Gag with RNAs with additional Gag mutants: '8N/WM', a Gag protein containing both the 8N and the WM mutations; 'SSHC', in which the first two Cys residues in each zinc finger are replaced with Ser; and '310', in which the four basic residues in the [29]RAPRKKG[35] linker between the two fingers are replaced with alanines.

The curve for binding of 8N Gag to both RNAs (red squares in *Figure 9A and B*) exhibits only one slope (2.0 and 2.8 for the dimeric Ψ and GRPE RNAs, respectively). This reflects the fact that we could not detect binding beyond the last data-point shown on the graph: evidently, there are really two slopes with 8N Gag, but the second slope is extremely steep. On the one hand, for the dimeric HIV Ψ RNA at NaCl concentrations between 200 and 500 mM, the difference in the curves for 8N Gag and WT Gag is very small. (In fact, at 500–550 mM NaCl, the $K_D$s for 8N Gag/dimeric Ψ are smaller than for WT Gag (see *Figure 9—figure supplement 3*)). On the other hand, the curve for 8N Gag/GRPE is shifted to higher $K_D$ values. Thus, decreasing the strength of MA-RNA interactions has a minor impact on binding to the dimeric Ψ, but a large effect on binding to the control RNA. This confirms that MA-RNA interactions make a significant contribution to the electrostatic component of binding.

As mentioned before, when Gag-Gag interactions were weakened (by means of ablating the dimer interface with the WM mutation), the presence of tRNAs revealed binding specificity for the dimeric HIV-1 Ψ (*Figure 6*). In a similar fashion, increasing the ionic strength reveals the high binding specificity of this protein for the dimeric HIV1-Ψ (compare the black circles in *Figure 9A versus 9B*); we could not detect binding of WM Gag to the GRPE RNA at NaCl concentrations higher than 200 mM, while the WM Gag/Ψ RNA curve closely resembles that for wild-type Gag. However, the steep slope appears at ≈ 300 mM rather than ≈ 400 mM NaCl. This shift of the salt concentration for the second slope is a surprising result, especially when considering that both the MA and NC domains, which are known to interact with RNA (*Shkriabai et al., 2006*), of this protein are identical to those of the wild-type protein. The lack of strong Gag-Gag interactions greatly decreased non-specific binding at salt concentrations slightly higher than physiological. In fact, comparing the plots for wild-

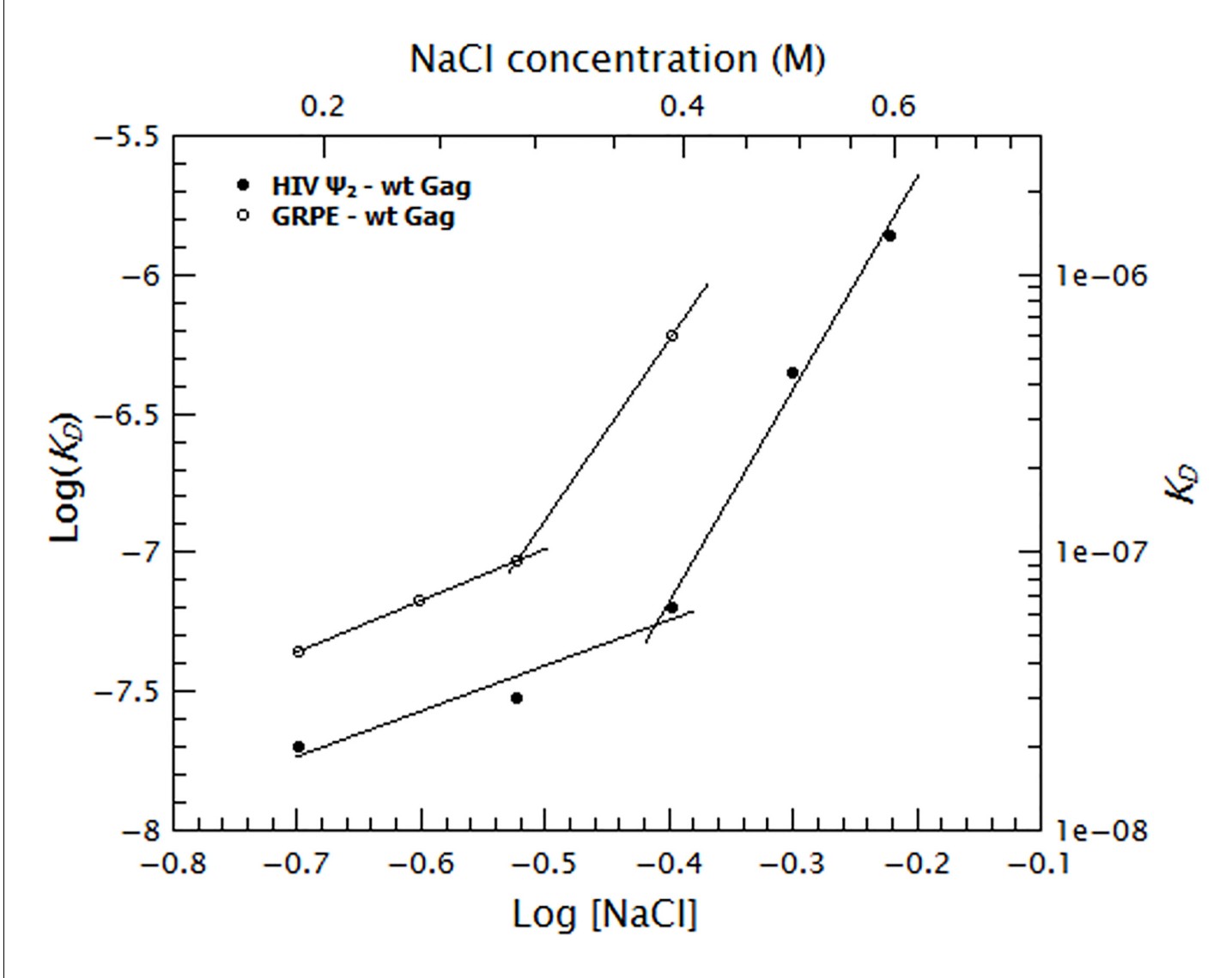

**Figure 8.** Binding of Gag to the dimeric HIV-1 Ψ has a stronger non-electrostatic component than to the GRPE RNA. The logarithms of the $K_D$s obtained in **Figure 7** were plotted as a function of logarithm of the NaCl concentration. The fact that these plots are not linear implies that there are two different binding modes at different salt concentrations. The intercept of each fit represents the dissociation constant in the absence of electrostatic interactions ($K_{D(1M)}$). At any salt concentration binding to the dimeric HIV-1 Ψ (black solid symbols) has a stronger non-electrostatic component than the GRPE RNA (black open symbols).

type, 8N, and WM Gag shows that Gag-Gag interactions have a stronger contribution to non-specific binding than MA-RNA interactions.

We also asked whether the specific interactions between Gag and RNA are strong enough to permit binding when MA-RNA and Gag-Gag interactions are both weakened. To do so we combined the 8N and WM mutations in one protein (8N/WM Gag). The green diamonds in **Figure 9A** show that binding of 8N/WM Gag to the dimeric Ψ exhibits the two-slope behavior. As expected, binding of this protein is considerably weaker than for the other proteins. In fact, we were not able to detect binding of this protein to the GRPE RNA at NaCl concentrations higher than 200 mM (see green diamonds in **Figure 9B**). These results show that simultaneously decreasing MA-RNA, Gag-Gag and the overall electrostatic interactions is not enough to abolish nanomolar binding to Ψ, while it profoundly depresses binding to GRPE RNA.

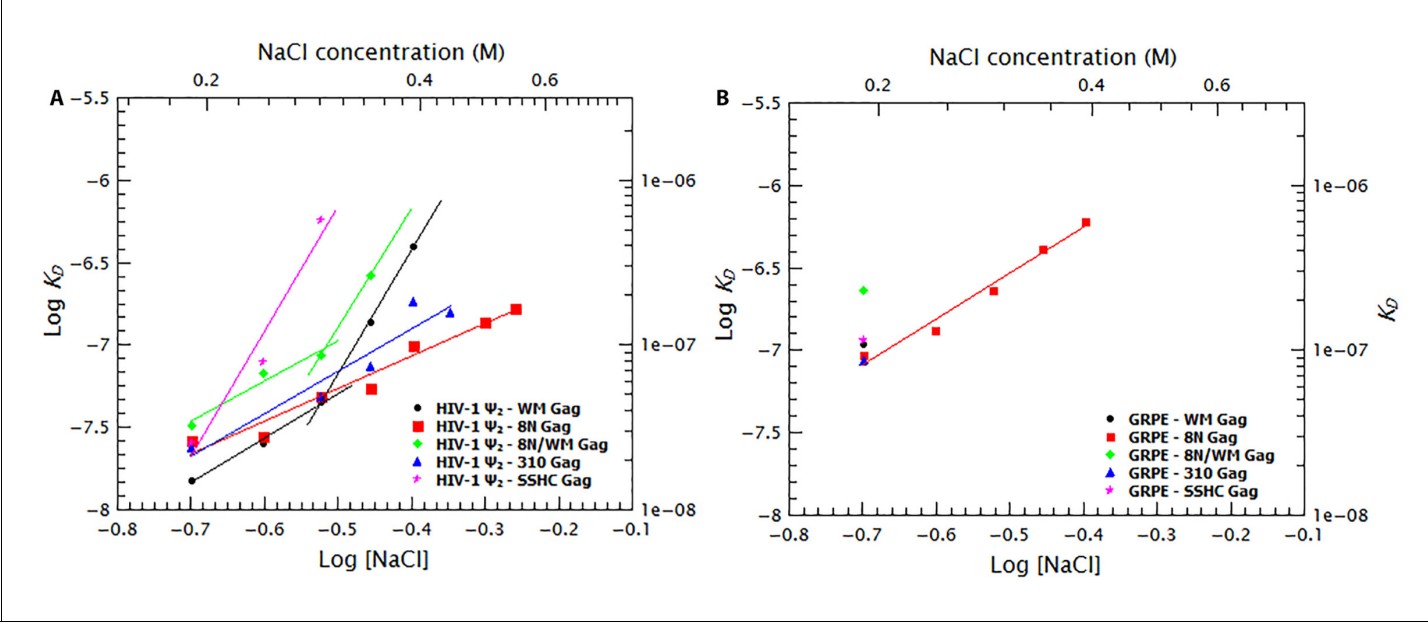

**Figure 9.** Binding to the GRPE RNA is driven by electrostatic interactions. Binding of WM Gag (black circles), 8N Gag (red squares), 8N/WM Gag (green diamonds), 310 Gag (blue triangles) and SSHC Gag (pink stars) to the dimeric HIV-1 Ψ (A) and to the GRPE RNA (B) was measured at different ionc strengths. (A) Decreasing MA-RNA (8N), Gag-Gag (WM), MA-RNA as well as Gag-Gag (8N/WM), as well as the electrostatic interactions between NC and the RNA (310), modulated the salt dependency of the interaction with dimeric HIV-1 Ψ. (However, SSHC binding still retains a strong electrostatic component). (B) Unlike the dimeric Ψ, the affinity of these proteins for GRPE RNA, with the exception of 8N Gag (red squares), is greatly impared by increasing salt concentration; we could not detect GRPE binding to these proteins at [NaCl]≥0.25 M. The $K_D$s calculated from *Figure 9—figure supplement 1B–F* and *Figure 9—figure supplement 2B–F* were used to generate panels A and B, respectively. A comparison of the plots of WT Gag (*Figure 8*) and 8N Gag (red squares in this *Figure 9*) is shown in *Figure 9—figure supplement 3*.

The following figure supplements are available for figure 9:

**Figure supplement 1.** Binding plots used to generate *Figures 8* and *9A*.

**Figure supplement 2.** Binding plots used to generate *Figures 8* and *9B*.

**Figure supplement 3.** Comparison of the salt dependency of WT and 8N Gag binding to the GRPE RNA and the dimeric HIV-1 Ψ RNA.

The two zinc fingers in the NC domain are essential for selection of the viral RNA in vivo (*Aldovini and Young, 1990*; *Gorelick et al., 1990*) and there is evidence that they are required for specific binding in vitro (*Fisher et al., 2006*; *Dannull et al., 1994*; *Berkowitz et al., 1993*). Thus, we tested their contribution to binding at different salt concentrations by using a Gag mutant in which the two zinc fingers were disrupted by mutating the first two Cys residues from each finger to Ser ('SSHC Gag'). These mutations destroy the ability of the NC domain to chelate zinc ions. The pink stars in *Figure 9A and B* show that at 200 mM NaCl the $K_D$s of SSHC Gag for the dimeric HIV-1 Ψ and GRPE RNA, respectively, are not significantly different from those of WT Gag. However, increasing the salt concentration drastically interfered with the binding to both RNAs. Binding of SSHC Gag to the dimeric Ψ RNA was undetectable at NaCl >300 mM (*Figure 9A*) and to GRPE RNA at NaCl ≥250 mM NaCl (*Figure 9B*). This salt-sensitivity demonstrates that the two zinc fingers are the major sources of non-electrostatic interactions (specific binding) and is fully consistent with the dramatic effects of zinc-finger mutations in vivo (*Aldovini and Young, 1990*; *Gorelick et al., 1990*). However, the fact that at 300 mM NaCl SSHC Gag binds detectably to the dimeric Ψ RNA, but not to the GRPE RNA, was somewhat unexpected. The implications of this result are considered in the discussion section.

To further analyze the contribution of the NC domain, we used a mutant protein (310 Gag) in which four positively charged amino acids between the two zinc fingers were mutated to alanines

($^{29}$RAPRKKG$^{35}$ of the NC domain to $^{29}$AAPAAAG$^{35}$). The blue triangles in *Figure 9A* show that binding of 310 Gag to the dimeric Ψ exhibited one slope; as we could not detect binding at 500 mM NaCl, the second slope is extremely steep. As expected, we could not detect binding of GRPE RNA to this protein at salt concentrations above 200 mM NaCl (blue triangle in *Figure 9*). The extremely weak binding, if any, of the GRPE RNA to this protein at NaCl concentrations above 200 mM NaCl shows that Gag-GRPE interactions are almost exclusively electrostatic, and that these interactions depend upon the basic linker between the zinc fingers in NC. In contrast, the robust binding of 310 Gag to Ψ demonstrates that Gag-Ψ interaction has a very large non-electrostatic component and that the linker is not crucial for this interaction.

## Discussion

Retroviruses such as HIV-1 selectively package their gRNA during virus assembly, despite the presence in the cell of a very large excess of cellular mRNAs, which can also be packaged (*Comas-Garcia et al., 2016*; *Rulli et al., 2007*; *Muriaux et al., 2001*). The selective packaging of gRNA is due to the presence of the 'packaging signal' or 'Ψ' in the gRNA. Several previous reports have indicated that the HIV-1 Gag protein binds preferentially to RNAs containing this signal in vitro (*Webb et al., 2013*; *Abd El-Wahab et al., 2014*; *Bernacchi et al., 2017*; *Dannull et al., 1994*; *Berkowitz et al., 1993*; *Damgaard et al., 1998*; *Geigenmüller and Linial, 1996*). In an effort to understand the mechanism of selective packaging of Ψ-containing RNAs, we have now performed a detailed analysis of the binding of recombinant HIV-1 Gag proteins to RNAs with or without Ψ.

Our findings can be very briefly summarized as follows. (1) binding of Gag to an RNA results in the physical collapse of the RNA and in quenching of the Cy5 fluorophore covalently linked to the 3' end of the RNA; (2) in all cases examined, Gag binds cooperatively to RNAs; (3) the affinities of Gag for all the RNAs tested are all very similar to each other at physiological ionic strengths; however, (4) specificity for Ψ is revealed when affinities are measured in the presence of an excess of a competing irrelevant RNA; when Gag-Gag and/or Gag-RNA interactions are impaired by mutations in Gag; or (as previously reported by Webb *et al.* [*Webb et al., 2013*]) when the ionic strength is increased. The results show that selective packaging of gRNA cannot be explained on the basis of the affinity of Gag for Ψ. However, the binding affinities represent the sum of both specific and non-specific interactions; when non-specific interactions are attenuated, it becomes clear that binding to Ψ has a far higher specific component than binding to other RNAs.

The results obtained with mutant Gag proteins and with competing tRNA are particularly informative. We found that addition of excess tRNA interferes significantly more with binding to the control RNAs (lacking Ψ) than to either monomeric and dimeric HIV-1 Ψ RNAs. This is in agreement with a previous study showing that addition of an excess of a competitor RNA helps to reveal binding specificity (*Berkowitz et al., 1993*). However, it has no detectable effect on the binding of 8N Gag (a mutant Gag in which basic residues in the MA domain have been replaced) to RNAs either with or without Ψ (*Figures 3* and *5B*). This shows that positively charged residues in MA play an important role in the binding to the non-Ψ RNAs and to tRNAs. This is consistent with the fact that the MA domain is mainly associated with tRNAs in the cytoplasm of virus-producing cells (*Kutluay et al., 2014*).

We found that the SSHC mutation, disrupting the zinc-chelating fingers in NC, drastically decreases specific binding to Ψ, in excellent concordance with mutational studies in vivo (*Aldovini and Young, 1990*; *Gorelick et al., 1990*). However, this mutant Gag still retains some residual preference for Ψ, as demonstrated both by its $K_D$ at 0.25 M NaCl and by the salt-resistance of the binding (*Figure 9*). Thus, preferential binding of Gag to Ψ cannot be entirely attributed to the two zinc fingers of NC. The residual specificity of SSHC Gag for Ψ is evidently undetectable by most experimental techniques and is obviously inadequate for selective packaging in vivo (*Aldovini and Young, 1990*; *Gorelick et al., 1990*). One possible explanation for this result is that the zinc fingers are not the only source of non-electrostatic interactions and that other domains of Gag contribute to specific Gag/Ψ interactions. Nonetheless, it is also possible that SSHC Gag only engages in electrostatic interactions, but the dimeric Ψ is a better substrate for electrostatic interactions than GRPE RNA; for example, its 3-dimensional structure might give it a higher charge density per unit volume than that of GRPE RNA.

Interestingly, a Gag mutant in which four basic residues between the two zinc finger have been replaced with alanines ('310' Gag) retained significant affinity for Ψ, but its ability to bind the GRPE control RNA was drastically curtailed, as indicated by the salt-sensitivity of this interaction (*Figure 9*). This salt-sensitivity suggests that just these four basic residues in NC (out of a total of 61 arginine and lysine residues in the Δp6 Gag used here) make an important contribution to the electrostatic interactions with RNA; these electrostatic effects are particularly important in binding to GRPE RNA. Perhaps, by virtue of their location, these four basic residues help to maintain the proper conformation of the NC domain or control the placement of RNAs on this domain. Mutation of these residues partially interferes with selective packaging of gRNA and eliminates viral infectivity in vivo (*Poon et al., 1996*; *Wu et al., 2014*).

In addition to the specific and non-specific interactions between Gag and RNA, the interactions between Gag molecules bound to an RNA molecule can also contribute to formation and stabilization of Gag/RNA complexes. The data with 'WM' mutant Gag, in which the Gag-Gag dimerization affinity in solution is reduced from $K_D$ ~10 μM to $K_D$ ~1 mM (*Datta et al., 2007b*), are especially striking. The binding of WM Gag to RNAs is still cooperative (*Figure 6A*), like that of WT and 8N Gag. Thus, Gag molecules, including those with the WM mutation, interact with each other when bound to RNA. It is important to note that in all of our titrations, the Gag concentrations were so low (generally ≤300 nM, always ≤600 nM) that Gag dimerization in solution is negligible (*Datta et al., 2007b*). The fact that the WM mutation does not reduce cooperativity suggests that the interactions between RNA-bound Gag molecules do not depend upon the dimer interface. While this is the only detectable interface for Gag-Gag interaction in solution, other interfaces (e.g., in the N-terminal domain of the CA domain (*Ganser-Pornillos et al., 2007*; *von Schwedler et al., 2003*; *Ganser-Pornillos et al., 2008*) and in SP1 (*Datta et al., 2011a*, *Datta et al., 2015*) must also be used in virus assembly. Hence, some of these interfaces could be exposed and contribute to cooperativity when Gag binds RNA in our experiments.

We also found that the binding of WM Gag to the control RNAs was more sensitive to addition of a competitor RNA or increase in salt concentration than that of WT Gag, while the binding of the mutant to Ψ-containing RNA was relatively resistant to these challenges (*Figures 6* and *9*; *Table 1*). These results suggest that the specific Gag/HIV-1 Ψ interactions are strong enough to overcome the destabilization caused by weakening the protein-protein interactions through the dimer interface. In turn, the data imply that Gag-Gag interaction *via* this interface significantly strengthens the binding of WT Gag to the control RNAs.

Addition of competitor tRNAs also revealed that binding of Gag to RNA exhibits hysteresis (*Figure 4*). We noted that hysteresis is greater with GRPE RNA than with monomeric HIV-1 Ψ RNA. This binding hysteresis is likely associated with the fact that re-equilibration of Gag/RNA complexes, upon addition of a second RNA, requires the dissociation of the initial complex. While binding is driven by Gag-nucleic acid interactions, dissociation of these complexes requires disrupting Gag-nucleic acid and Gag-Gag interactions. This implies that Gag re-equilibration, after addition of a second RNA species, must overcome the energetic barrier resulting from breaking Gag-Gag interactions. Therefore, the smaller degree of binding hysteresis for the HIV-1 Ψ RNA, compared to the GRPE RNA, implies that the energetic barrier for re-equilibration is lower in the presence of specific Gag-Ψ interactions.

The ability of a protein to collapse an RNA (i.e., reduce its hydrodynamic radius and increase its diffusion constant; see *Figure 1*) has previously been reported with capsid proteins of other viruses (*Borodavka et al., 2012*, *2013*). Presumably, it occurs when the binding of positively charged regions of the protein to the RNA backbone reduces the electrostatic repulsion within the RNA; this activity very likely comes into play during assembly of the virus particle. The fact that the WM mutation profoundly reduces the ability of HIV-1 Gag protein to collapse RNAs (*Figure 6—figure supplement 1*), as well as the cooperativity of RNA collapse by WT and 8N Gag proteins (*Figures 1B* and *5C*), implies that protein-protein interaction (through the dimer interface) makes an important contribution to RNA collapse. The dependence of RNA collapse on protein-protein interactions is also consistent with recently described molecular dynamics simulations on RNA collapse by capsid proteins (*Perlmutter and Hagan, 2015*). It was interesting to note that the presence of excess yeast tRNA greatly reduced the magnitude of RNA collapse (*Figure 3—figure supplement 1*). This is even true with 8N Gag (*Figure 5*), for which RNA binding per se is almost unaffected by tRNA

addition (*Figure 5B*). These observations suggest that tRNA is incorporated into the complexes of Gag and tagged RNA, reducing compaction, *via* one or more Gag domains other than MA.

Considered together, all of these results point to the following conclusions. The affinity of a Gag molecule for an RNA represents the sum of (I) non-electrostatic interactions; (II) electrostatic interactions; and (III) the interactions between two or more Gag molecules bound to a single RNA molecule. Interactions of the first type, mediated largely by the NC zinc fingers, impart specificity for $\Psi$. To a first approximation, the electrostatic interactions are presumably similar for different RNAs; our data show that the basic residues in the MA domain contribute significantly to these interactions. It seems likely that the basic character of the NC domain also plays a role in these interactions. The specificity for $\Psi$ is revealed in the presence of mutations that reduce the strength of either the second or third types of interaction, or by addition of RNAs that compete for these interactions, or in an ionic environment that screens the second type of interaction.

Comparison of the effects of added tRNAs or increased ionic strength upon the apparent affinities of WT and 8N Gag for the labeled RNAs (*Figures 2B* and *3 vs. 5B*, and *Figures 8 vs. 9*) shows that binding to $\Psi$ and non-$\Psi$ RNAs involves both MA and NC domains. This result does not appear to be consistent with a previous binding model in which binding of Gag to $\Psi$ was suggested to involve only NC, while binding to a non-$\Psi$ RNA is driven by interactions with both MA and NC (*Webb et al., 2013*). Webb and co-workers measured Gag/RNA equilibria by doing salt titrations of pre-mixed Gag/RNA complexes (at a fixed Gag:RNA ratio), while we titrated a fixed amount of RNA with varying concentrations of Gag at different salt concentrations. Presumably, the differences between the experimental approaches are responsible for the resulting discrepancy in binding models. It is somewhat surprising that the apparent affinity of 8N Gag for the dimeric HIV-1 $\Psi$ between 500 and 550 mM NaCl is greater than that of WT Gag (*Figure 9—figure supplement 3*). However, the interaction of the WT MA domain with RNAs is presumably almost exclusively electrostatic and is thus negligible at these high salt concentrations; we surmise that under these conditions, repulsion between positively charged WT MA domains of two or more RNA-bound Gag molecules raises the energetic cost of RNA-binding, and that this repulsion is reduced by the 8N mutations.

Previous studies on protein-nucleic acid binding show that the sensitivity of the binding to an increase in ionic strength is a direct reflection of the net number of charges participating in the binding ($Z_{eff}$) (*Webb et al., 2013*; *Record et al., 1978*, *1976*; *Rouzina and Bloomfield, 1997*; *Rye-McCurdy et al., 2015*); in these experiments, plots of the logarithm of the $K_D$ vs. logarithm of the monovalent ion concentration give a straight line, whose slope represents the $Z_{eff}$. However, in our experiments, these plots were bent rather than straight (*Figures 8* and *9*). According to the above mentioned studies on protein-nucleic acid interactions, the bent shape would imply that the $Z_{eff}$ is different under the different ionic regimes. It seems likely that the change in slope reflects a conformational change in the Gag protein, which is known to assume very different conformations under different conditions (*Datta et al., 2007a*, *2011b*). Remarkably, the salt concentration at which this hypothetical change occurs is evidently determined in part by the identity of the RNA.

It might be proposed that the curves are really straight, and that our measurements in the lower salt concentrations are inaccurate at the very high affinities seen with WT Gag and $\Psi$ RNA. In fact, the RNA concentration in these experiments (15 nM) is near the measured $K_D$s, and thus we cannot completely exclude the possibility that the actual $K_D$ is even lower than our results indicate. However, it is important to note that the curve for WT Gag and GRPE RNA is also bent (*Figure 8*), although the binding at each salt concentration is somewhat weaker than with $\Psi$ RNA and we have no reason to doubt the accuracy of these measurements. Several other bent curves were also found (*Figure 9A*). Furthermore, the fact that we were not able to measure a $K_D$ after the last plotted point for 8N Gag/dimeric $\Psi$, 8N Gag/GRPE and 310 Gag/dimeric $\Psi$ indicates that there is an abrupt change in the $K_D$ at increased salt concentrations. Such an abrupt change is only possible if the curves for those protein and RNA combinations are bent.

It is particularly surprising that the curves exhibit steeper slopes at higher ionic strengths than at low ionic strength. One might naively expect that as the salt concentration is increased the electrostatic component decreases, ultimately reaching a point at which only the non-electrostatic component remains significant. This scenario would produce a curve with a steep slope (i.e., high salt sensitivity) at low salt and a shallow one at high salt; however, our plots (*Figures 8* and *9*) show the reverse pattern. As we have mentioned before, the zinc fingers are primarily responsible for non-electrostatic binding (*Figure 9*). Perhaps the change in slope reflects a change in the conformation

of the NC domain, which in turn alters the non-electrostatic interactions of this domain. *Figure 10* shows that the 'upward bent' curve for salt titration of WT Gag/dimeric Ψ (red line and circles) can be generated by using a model (blue line) in which the $K_D$ is a weighted sum of the electrostatic interactions (black line and circles) and strong non-electrostatic interactions (solid green line). In the first binding mode (I) the conformation of Gag is such that binding is mostly non-electrostatic. In the second binding mode (II), the salt-induced conformational change in Gag gradually decreases the contribution of the non-electrostatic interactions (dotted green line). (The details of this model are explained in the supplementary information section.) Again, with 8N Gag/dimeric Ψ, 8N Gag/GRPE, and 310 Gag/dimeric Ψ, only a single slope is visible: this is because we could not detect binding at higher ionic strengths (0.6, 0.45 and 0.5 M NaCl, respectively). The $K_D$s at higher salt concentrations in these cases must be >1 μM. In other words, these curves do have a second slope, but this slope is so steep that we could not make measurements at these high salt concentrations.

As noted above, the distinctive salt resistance of the binding of Gag to Ψ was previously reported by Webb *et al.* (*Webb et al., 2013*). Their experimental approach was somewhat different from ours; they could not have detected the bend in these curves, as the model to which they fitted their salt-titration data assumed a single slope. Interestingly, bent curves were also observed by Record *et al.* (*Record et al., 1977*), although their experimental system was different from ours in several ways.

While the majority of our experiments compared binding to dimeric Ψ RNA (the substrate for selective packaging in vivo) with binding to GRPE RNA, we also characterized binding to monomeric Ψ and to the MoMLV Ψ (RNAs containing MoMLV Ψ are not selectively packaged by HIV-1 Gag

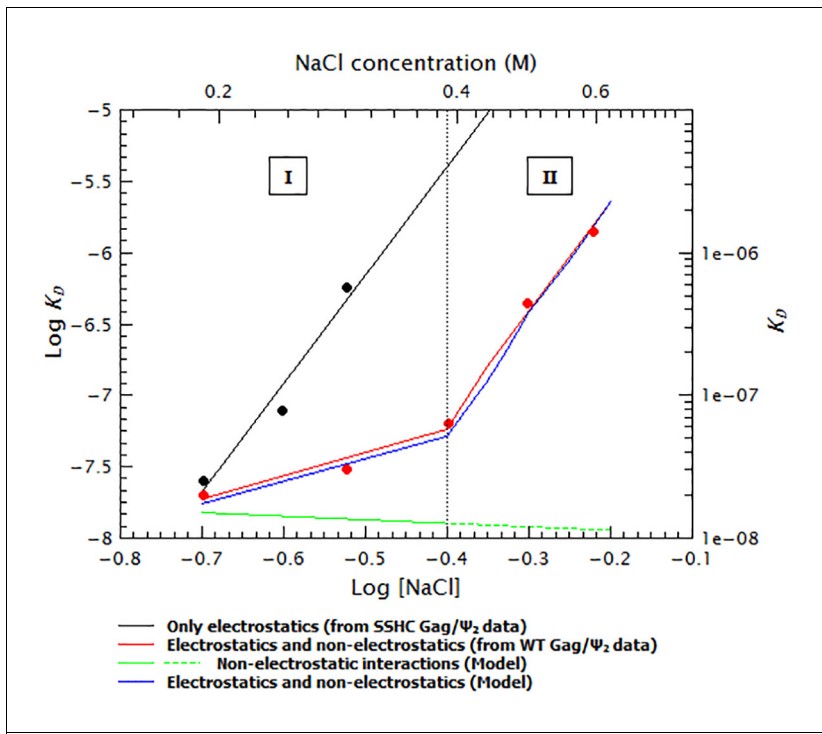

**Figure 10.** The 'upwards bent' salt titration indicates that binding of Gag to RNAs has a very strong non-electrostatic component. The concave or 'upwards bent' shape of the salt titration data for WT Gag/Ψ₂ (red circles) was obtained by using a model in which the observed $K_D$ (blue line) is the weighted sum of strong non-electrostatic (green line) and electrostatic (black line) interactions. The salt titration data for SSHC Gag/Ψ₂ (black circles) was used to extract the pure electrostatic component of binding to the dimeric HIV-1 Ψ (black line) and the WT Gag/Ψ₂ represents the sum of both electrostatic and non-electrostatic interactions (red line). In the first binding mode (I), binding is driven by non-electrostatic interactions (solid green line). In the second binding mode (II), a conformational change in Gag gradually decreases the relative contribution of the non-electrostatic interactions (broken green line).

protein). In general, our data shows that binding of these two RNAs to Gag and its mutants is intermediate between the dimeric HIV-1 Ψ and the GRPE RNA. Under most experimental conditions the binding affinity of WT, as well as of most mutant Gag proteins, for the dimeric and monomeric HIV-1 Ψ were very similar to each other. Nonetheless, the dimeric RNA always exhibited a higher binding affinity than its monomeric counterpart. (Interestingly, the binding affinity for the dimeric HIV-1 Ψ was considerably higher than that for the monomeric Ψ in the tRNA competition experiments with WM Gag.) In fact, because the difference in the overall binding character between the monomeric and dimeric HIV-1 Ψ were small, we decided not to test the effects of RNA dimerization on the MoMLV Ψ. It was recently reported (*Kharytonchyk et al., 2016*) that the precise start site of genomic RNA affects its ability to dimerize; our data suggest that the specific interaction of HIV-1 Gag with HIV-1 Ψ is enhanced by dimerization of the RNA, and would thus, in a genomic RNA molecule, be influenced by the base at which synthesis of a given RNA molecule was initiated.

What can these results tell us about the mechanism of selective gRNA packaging in vivo? Taken together, the data show that while Gag has similar affinities for different RNAs at near-physiological ionic strength, there are striking differences in the character of the binding: interaction with Ψ-containing RNA has a much larger specific, non-electrostatic component than that with the control RNAs. In addition, the cellular environment could certainly affect Gag-RNA and Gag-Gag interactions in profound ways (*Becker and Sherer, 2017*; *Lingappa et al., 2014*), thus allowing for highly specific interactions with the gRNA. We have, however, suggested that Ψ-specific interactions could lower the energetic barrier required to nucleate particle assembly more efficiently than binding to other RNAs (*Comas-Garcia et al., 2016*). This would explain selective packaging and is also consistent with the fact that in the absence of the gRNA, Gag assembles into particles and packages cellular RNAs. It is also consonant with data implying that the dimeric HIV-1 Ψ supports virion assembly more efficiently than cellular RNAs in vivo (*Nikolaitchik et al., 2013*). Results consistent with this hypothesis were also recently reported by Carlson et al. (*Carlson et al., 2016*) and by Dilley et al. (*Dilley et al., 2017*).

## Materials and methods

### Plasmid construction

The HIV-1 Ψ (nt. 193–368) and the Gag-Pol Frame Shift (GRPE) (nt. 2004–2179) sequences were amplified by PCR from a pNL4-3 plasmid. The Moloney Murine Leukemia Virus Ψ (MoMLV Ψ) sequence (nt. 202–377) was amplified from a proviral plasmid (GenBank J02255.1). A T7 promoter and an *NcoI* restriction digestion site were inserted at the beginning and end of the viral sequence, respectively. The PCR fragments were inserted into a Topo vector (ThermoFisher Scientific, Waltham, MA USA), by Topo cloning. To increase the labeling density of the RNAs (see below) a 15-base polyA tail was added between the end of each viral sequence and the *NcoI* restriction site.

### RNA in vitro transcription and RNA labeling

Before transcription, the plasmids were digested with *NcoI* and *BglII* and the fragment containing the viral sequence was purified from a 1% agarose/TAE gel by using a Nucleo spin kit (Macherey-Nagel, Düren, Germany) and stored at −20°C. In vitro transcription of the fragments was done with a MEGAshortscript kit (ThermoFisher Scientific) following the vendor's instructions. The RNA was purified by gel electrophoresis (6% polyacrylamide/TBE-UREA gel) and eluted overnight in a Thermomixer (Eppendorf, Hamburg, Germany) at 30°C and 1000 rpm (50 mM Tris-HCl pH 7.5, 10 mM EDTA, 300 mM NaCl and 0.1% SDS). The RNAs were further purified by phenol:chloroform extraction and ethanol precipitation (*Green and Sambrook, 2012*). RNA concentrations were measured in a UV-Vis NanoDrop 1000 and their integrity was confirmed by gel electrophoresis (6% polyacrylamide/TBE-UREA). The RNAs were 3'-end Cy5 labeled by ligating pCp-Cy5 (Jena Bioscience GmbH, Jena, Germany) to the 3'-end of the RNA with T4 RNA ligase (New England Biolabs, Ipswich, MA USA) (*Wang et al., 2007*). To purify the RNAs the samples were run through three consecutive illustra microspin G-50 columns (GE Healthcare Life Sciences, Pittsburgh, PA USA) per 35 µL of labeling reaction, concentrated with an RNA clean and concentrator-25 kit (Zymo Research, Irvine, CA USA) and eluted with double-autoclaved double deionized water (dd-Water). The labeling yield was determined according to the manufacturer's instructions with a UV-Vis NanoDrop 1000 and the

integrity of the RNAs was determined by gel electrophoresis (6% polyacrylamide/TBE-UREA). The labeling yield was between 0.2 and 0.35. *Figure 1—figure supplement 1a* shows two characteristic denaturing gels for the purified labeled RNAs (6% polyacrylamide/TBE-UREA gel). These gels show that the in vitro transcripts are intact, without significant levels of early-termination products or degradation products. *Figure 1—figure supplement 1b* is a native gel (3% MetaPhor Agarose [Lonza, Basel, Switzerland] in TAE) for the dimeric HIV-1 Ψ RNA, showing that the RNA is not aggregating significantly, even after being incubated for dimerization. The RNAs were stored in water at −80°C.

## Protein expression and purification

All proteins, without any affinity tags, were expressed in BL21(De3)pLysS *E. coli*. HIV-1 Δp6 Gag (from here on called *WT-Gag*) (*Campbell and Rein, 1999*), Δp6 WM Gag (WM Gag) (*Datta et al., 2007b*), Δp6 310 Gag and Δp6 SSHC-Gag were purified following a previously established protocol (*Datta and Rein, 2009*). Δp6 310 Gag was generated by Quikchange (Agilent Genomics, Santa Clara, CA USA) site-directed mutagenesis changing the $^{29}$RAPRKKG$^{35}$ between the two zinc fingers in the NC domain of Δp6 Gag to $^{29}$AAPAAAG$^{35}$, while for the Δp6 SSHC Gag mutant, the sequences $^{14}$KCFNCG$^{19}$ and $^{35}$GCWKCG$^{40}$ within the two zinc fingers in NC were changed to $^{14}$KSFNSG$^{19}$ and $^{35}$GSWKSG$^{40}$. Purification of Δp6 8N Gag (*Zhou et al., 1994*) (8N Gag) required the following modifications; (i) bacterial cultures were grown at 30°C, (ii) IPTG was added when the O.D. at 600 nm was 0.4, (iii) the bacteria were induced for 2 hr, (iv) the lysis buffer composition was modified by adding 1 mg/mL of lysozyme, 2.8 units/mL of DNase I, 23 μg/mL of RNaseA, 5 mM MgCl$_2$ and 1 mL of 100x Halt protease (Thermo Fisher Scientific), (v) to remove any nucleic acids a 5% polyethylamine (PEI) solution was added to a final concentration of 0.05% after the ammonium sulfate step (see protocol in [*Datta and Rein, 2009*]). The sample was centrifuged at 12,000 RCF for 15 min and the supernatant was used for the subsequent ammonium sulfate precipitation, and *vi*) at the phosphocellulose resin step the protein was eluted after the 0.2 M NaCl washes with a 1.0 M NaCl buffer. We combined the 8N and WM mutants by site directed mutagenesis of the 8N-Gag plasmid; 184W:185M to 184A:185A in the CA domain (Δp6 8N/WM Gag). 8N/WM Gag purification was purified following the same modified protocol as 8N Gag; however, before size exclusion chromatography the sample was subjected to sequential anion and cation exchange chromatography on tandem Q- and SP-Sepharose columns (GE Healthcare), respectively. In all cases, the proteins were further purified by size-exclusion chromatography with a Superose 12 10/300 GL column (GE Healthcare Life Sciences) in an Äkta Purifier FPLC (GE Healthcare Life Sciences). The purity of the eluted proteins was confirmed by SDS-PAGE and coomassie staining and the concentration was estimated by measuring the absorbance at 280 nm. All proteins were kept in 20 mM Tris pH 7.5, 0.5 M NaCl, 1 μM ZnCl$_2$, 0.1 mM Phenymethylsulfonyl fluoride (PMSF), 1 mM $\beta$−Mercaptoethanol ($\beta$ME) and 10% glycerol buffer and stored at −80°C in ≈5 μL aliquots at ≈20–40 μM. To determine if the Δp6 Gag could form aggregates after freezing/thawing cycles we labeled Δp6 Gag with Alexa Fluor 647 by adapting a previously developed protocol to determine its hydrodynamic radius by FCS (see FCS methods section) (*Comas-Garcia et al., 2014*). We also tested for the presence of aggregates in Δp6 Gag that had been subjected to freezing/thawing cycles by sedimentation velocity measurements. Briefly, solutions of Δp6 Gag in 20 mM Tris HCl, pH 7.4, 0.5 M NaCl, 1 μM ZnCl$_2$, 1 mM TCEP were prepared at concentrations of 2 μM (a concentration in which this protein is in monomeric form) by dilution from a stock at ~20 μM. The solution density and viscosity, and the partial specific volume of Gag protein at 20°C were estimated using SEDNTERP (*Cole et al., 2008*). Sedimentation velocity experiments were conducted in a Beckman Coulter Proteome Lab XL-I analytical ultracentrifuge (Brea, CA USA) at 45,000 rpm and 20°C. Sedimentation of Gag in two channel Epon centerpiece cells were monitored by both absorbance (280 nm) and Raleigh interference (665 nm). Time-corrected data (*Zhao et al., 2013*) were initially analyzed in SEDFIT 14.4 f (*Schuck, 2000*) as a continuous c(s) distribution covering an s range of 0.0–100 s with a resolution of 100 and a confidence level (F-ratio) of 0.95. Excellent fits were obtained with r.m.s.d. values between 0.003 and 0.004 absorbance units. The c(s) profiles, consistent with a reversible monomer–dimer self-association, shows a single peak centered at 3.25 s (S$_{20, W}$ of 3.6 s), with >~97% of all signal between 0.5 to 8.0 s.

## RNA dimerization

The HIV-1 Ψ dimerization protocol was based on a previously published method (*Laughrea and Jetté, 1997*). Briefly, the RNA (≈1 μM in dd-water), was heated at 92°C for 5 min, cooled-down on ice for 5 min, then 1/5 vol of a 5X dimerization buffer (250 mM Tris-HCl pH 7.5, 1.5 M KCl and 25 mM $MgCl_2$) was added, followed by an incubation on ice for 30 min and then at 55°C for 15 min. Finally, the RNA was diluted with the Binding Buffer (20 mM Tris-HCl 7.5, 0.2 M NaCl, 5 mM $MgCl_2$, 1 μM $ZnCl_2$, 0.1 mM PMSF, 1 mM βME and 0.05% (v/v) Tween 20) to a final concentration of 0.03–0.01 μM. The monomeric HIV-1 Ψ was generated by diluting the Ψ RNA from ≈1–2 μM to 0.03–0.01 μM with binding buffer, followed by thermally annealing; 85°C for 1 min, cool down to 4°C at a rate of 0.1 °C/s, 4°C for 3 min and 37°C for 15 min. This protocol was also used to thermally anneal the GRPE and MoMLV Ψ RNAs.

## Gag/RNA binding reactions

All reagents and chemicals were RNase and DNase free and all solutions were sterilized. The binding reactions were performed in RNase and DNase free, freshly autoclaved 1.5 mL Eppendorf tubes (low-binding); the final volume was 22 μL. The Binding Buffer was freshly made just before use. The proteins were thawed on ice and binding buffer was added to reach a final Gag concentration of 1.1 μM. For the competition experiments, the yeast tRNAs (Sigma-Aldrich, Stl Louis, MO USA) were incubated with the protein for about 10–20 min before the labeled RNAs were added (unless indicated otherwise). In all cases the tRNAs were added in a 50:1 (wt/wt) tRNA:RNA ratio. In all experiments the final concentration of the viral-derived RNAs was 15 nM. The samples were protected from the light whenever possible. The reactions were incubated in the dark overnight at 4°C. Each plot represents one experiment, but each experiment was performed at least twice. In the diffusion plots, the symbols and error bars represent the mean and standard deviation of 10 measurements.

## Fluorescence correlation spectroscopy (FCS) and data analysis

All FCS measurements were done in a custom-made dual-color FCS instrument built around an inverted Olympus IX71 microscope with a UPLSAPO water immersion objective (60X, 1.2 NA, Olympus Corp. of Americas, Center Valley, PA USA) and with custom-designed optics. The emission fluorescence was detected using Single Photon Counting Module (SPCM-AQRH-16-FC, Excelitas Tech. Corp., Waltham, MA USA) through a multimode optical fiber that acted as a pin-hole. The Transistor-Transistor logic signal (TTL) high output from the SPCMs was collected using a 32-bit, 100Mhz counter card (PCIe-6323, National Instruments, Austin, TX, USA). An in-house LabView based control and acquisition interface was used for timestamping the photons and intensity trajectories were analyzed off-line to generate the autocorrelation curves. For the FCS measurements a 5 μL aliquot was placed between two 30 mm round glass slides (No. 1.5) and sealed with a 1 mm silicone sheet. All measurements were done at room temperature. Before starting the measurements, the FCS setup was calibrated by using a solution containing 10 nM Alexa Fluor 647. There are several diffusion coefficients reported for Alexa Fluor 647; therefore, we measured the diffusion coefficient of a AF647-labeled protein with known diffusion coefficient (lysozyme) to estimate the proper diffusion coefficient of the dye (see *Supplementary file 1*). Based on these results we used a value of 300 $\mu m^2$/s for the diffusion coefficient of Alexa Fluor 647 to calculate the confocal volume. It should be pointed out that both of the values for the diffusion of dye gave diffusion coefficients for the lysozyme that are greater than the reported values (see *Supplementary file 1*). This implies that the diffusion coefficients that we are reporting for the RNA and Gag/RNA complexes are somewhat larger than they should be. The data was acquired for 10–20 s and each sample was measured at least 10 times. The data was binned at 1 MHz and the autocorrelation curves were fitted using the QtiPlot 0.9.9-rc15 software (Ion Vassilief, ProIndep Serv, Craiova, Romania). The details of the mathematical models used to describe the autocorrelation curves and to generate and fit the binding plots are in the supplementary information. *Figure 1—figure supplement 2b* shows the normalized autocorrelation function of Alexa Fluor 488-labeled HIV-1 WT Δp6 Gag (at 300 nM). The diffusion coefficient for Gag measured by FCS is ≈106 $\mu m^2$/s, thus by using the Stokes-Einstein equation (which assumes a spherical volume) we obtained a hydrodynamic radius ($R_h$) ≈2.5 nm. Further, *Figure 1—figure supplement 3* shows the

normalized autocorrelation functions for the monomeric and dimeric HIV-1 Ψ, GRPE and MoMLV Ψ RNAs (panels A, B, C and D, respectively). The fact that the value of these autocorrelation functions, at lag times where large aggregates are to be expected (lag times between 0.01 and 0.1 s) is zero indicates that at 15 nM these RNAs do not form irreversible aggregates.

## Supplementary information

### Fluorescence correlation spectroscopy and data analysis

The autocorrelation function, $G(\tau)$, can be written as:

$$G(\tau) = G(\tau)_{diffusion} * \chi(\tau)_{kinetics} \tag{1}$$

The photo-physical properties of the fluorophores contribute to fluorescence fluctuations on time scales much faster than the characteristic diffusion time of macromolecules, $\tau_D$, $(\tau_t <_D)$. These fluctuations are related to the formation of a triplet state during excitation of the fluorophore:

$$\chi(\tau)_{kinetics} = 1 - T + T * e^{-\tau/\tau_t} \tag{2}$$

where $T$ and $\tau_t$ are the fraction and characteristic time of fluorophores in their triplet state, respectively. This equation was used in the autocorrelation function (*Equation 1*) to fit the autocorrelation curves of the Alexa Fluor 647 dye for calibration purposes.

Cy5 also undergoes a reversible photon-induced isomerization process that produces a non-fluorescent state.

$$\chi(\tau)_{kinetics} = 1 - T + T * e^{-\tau/\tau_t} - B + B * e^{-\tau/\tau_B} \tag{3}$$

where $B$ is the fraction of molecules that undergo this isomerization process and $\tau_B$ is the characteristic time describing the conversion rate of this process ($\tau_t < t_B$). This equation was used in the autocorrelation function (*Equation 1*) to fit all the correlation curves of the Cy5-labeled RNAs.

The diffusion of a macromolecule in a 3D Gaussian-shaped confocal volume can be described by the following equation

$$G(\tau)_D = \langle N^{-1} \rangle \left(1 + \frac{\tau}{\tau_D}\right)^{-1} \left(1 + \frac{\tau}{S^2 * \tau_D}\right)^{-1/2} \tag{4}$$

here $<N^{-1}>$ is the average number of particles in the confocal volume, $S$ is a geometric factor of the confocal volume ($S = 5$ for our setup), $\tau_D$ is the characteristic diffusion time, which depends on the diffusion coefficient, $D$, and the beam waist, $w_{xy}$ in the x,y-dimensions of the laser focus:

$$\tau_D = \omega_{xy}^2/4D \tag{5}$$

The autocorrelation curves with Alexa dyes were fitted by using a model with one triplet state (*Equation 1* and *Equation 2*), while the data with Cy5 dyes (RNA) were fitted by adding a term that takes into account the rate of isomerization between the dark and fluorescent isomer of Cy5 (*Equation 1* and *Equation 3*). In all cases the data was fitted by using a non-linear scaled Levenberg-Marquardt algorithm.

To calculate the degree of cooperativity of RNA collapse the diffusion coefficient of the RNAs at each point of the Gag titrations ($D_{[Gag]}$) was normalized relative to that of a control sample of pure RNA ($D_{[0]}$) (measured for every titration); the normalized diffusion coefficient of the RNAs, $D_N$, in the presence of Gag becomes zero as it approaches $D_{[0]}$:

$$D_N = \frac{D_{[Gag]}}{D_{[0]}} - 1 \tag{6}$$

Plotting $D_N$ as a function of Gag concentration results in a sigmoidal curve, hence we derived the following equation to fit these plots:

$$D_N = \frac{D_{[Gag]}}{D_{[0]}} - 1 = \frac{D_{max}}{1 + \left(\frac{K}{[Gag]}\right)^n} \tag{7}$$

where $D_{max}$ is the maximum degree of RNA collapse in the titrations and $K$ is the concentration of protein required to produced $D_{max}/2$. n is a fitting coefficient which is analogous to the Hill coefficient in the Hill equation. For the fitting routines $D_{max}$ was kept constant while n and K were freely fit.

The quenching data was converted into fraction of bound RNA to generate binding plots by using the following formula:

$$f([Gag]) = \frac{I_0 - I([Gag])}{I_o - I_p} \qquad (8)$$

where $I_0$ is the fluorescence intensity of the RNA in the absence of Gag, $I([Gag])$I is the fluorescence intensity of the RNA at any given Gag concentration and $I_p$ is the fluorescence intensity at the plateau of the quenching curves.

The binding plots were fitted by using a cooperative binding model (Hill equation):

$$f([Gag]) = \frac{1}{1 + \left(\frac{K_D}{[Gag]}\right)^{n_H}} \qquad (9)$$

were $n_H$ is the Hill coefficient, $K_D$ is the nanomolar concentration at which half of the RNA is saturated with Gag (also called dissociation constant) and $[Gag]$ is the nanomolar concentration of Gag. To test the validity of this binding mechanism we also fitted the data with a multiple binding non-cooperative model and with a one-to-one binding model, respectively:

$$f([Gag]) = \frac{n}{1 + \left(\frac{K_D}{[Gag]}\right)} \qquad (10)$$

$$f([Gag]) = \frac{1}{1 + \left(\frac{K_D}{[Gag]}\right)} \qquad (11)$$

where n is the number of binding sites that are occupied with Gag molecules.

The log $K_D$ vs Log [NaCl] plots were fitted by doing a least squares linear fit using the following formula:

$$logK_D = logK_{D(1M)} + Z_{eff}^{\cdot} \log[Na^+] \qquad (12)$$

In most instances the data was not linear over the entire salt concentration regime, thus we used two independent linear equations to fit the data. These two equations were used to obtain the fitting parameters for the shallow and steep slopes.

It should be noted that, because of the nature of the interaction of protein with the fluorophore, the quenching is all-or-none. That is, as mentioned in the main text, excitation of Cy5 induces a reversible conversion to a completely non-fluorescent state, evidently stabilized by steric interactions in the Gag-RNA complexes. This is distinct from the quenching experienced by many fluorophores, in which there is a change in emission spectra or the relaxation time of the dye (*Levitus and Ranjit, 2011*; *Stennett et al., 2014*). Since the conversion is all-or-none, there is a direct, linear relationship between the number of molecules quenched and the decrease in total fluorescence. Moreover, quenching eventually reaches a plateau, where further addition of Gag protein has no further effect on fluorescence; we interpret this to mean that at this plateau, all the RNA molecules are bound by one or more Gag molecules. This stoichiometry is taken as 100% bound RNA in the Gag titrations. Although these are the only assumptions we used in converting the quenching data into binding plots, it should be noted that a model positing cooperative binding gave a far better fit to the binding data than other models (*Figure 2 – figure supplementary 1*).

## Binding model of the salt titrations

To explain the concave or 'upward bent' shape of these salt titrations (*Figures 8* and *9*) we generated a simple model (blue line in *Figure 10*) in which the salt titration data for SSHC Gag/$\Psi_2$ (black circles) was used to approximate the pure electrostatic component of binding of WT Gag to the

dimeric $\Psi$ (black line). The salt titration data for WT Gag/$\Psi_2$ (red circles) was used to represent the sum of both electrostatic and non-electrostatic interactions (red line).

This model assumes that there are two binding modes. In the first (lower salt) binding mode (I) the interaction between WT Gag and the dimeric $\Psi$ is mostly non-electrostatic (solid green line). This assumption is based on the facts that between 0.2 and 0.4 M NaCl the salt-sensitivity of binding is minimal and that disruption of the zinc fingers (in SSHC Gag) greatly increases the salt sensitivity of binding. In the second binding mode (II) we assumed that the salt-dependent conformational change in Gag gradually reduces the strength of the non-electrostatic interaction (broken green line). This assumption is based on the high salt-sensitivity of this second binding mode.

To model the experimental data, measured affinity at each salt concentration was taken to be the sum of the electrostatic component (black line, taken from SSHC Gag binding) and a hypothetical non-electrostatic interaction (green line). In the first binding mode (I), the weighted sum of these two interactions was adjusted so that it produced the same slope and intercept as the experimental data; the weight of the non-electrostatic interactions was greater than the electrostatic one and both contributions were kept constant. In the second binding mode (II), the weight of the non-electrostatic interactions was reduced. The ratio of each interaction with respect of each other was chosen to obtain a slope and intercept similar to the experimental data.

It is important to point out that the goal of this model is not to fit the experimental data but to show that the concave or 'upward bent' shape of the salt-titrations indicates that binding of Gag to RNAs has a significant non-electrostatic component and that the strength of this interaction varies as the salt concentration is increased.

## Acknowledgements

This study was supported by the Intramural Research Program of the NIH, National Cancer Institute, Center for Cancer Research; in part with funds from the Intramural AIDS Targeted Antiviral Therapy program; and in part with Federal funds from the Frederick National Laboratory for Cancer Research, National Institutes of Health, under contract HHSN26120080001E. The content of this publication does not necessarily reflects the views or policies of the Department of Health and Human Services, nor does mention of trade names, commercial products, or organizations imply endorsements by the U.S. Government.

We thank Demetria Harvin for her technical assistance. We gratefully acknowledge the help of Martin Meier-Schellerheim of the National Institute of Allergy and Infectious Diseases with access to the dual-FCS setup. We also thank Michael Hagan, Stephen Lockett, Karin Musier-Forsyth, Ioulia Rouzina, and Roya Zandi for their critical discussions.

## Additional information

### Funding

| Funder | Grant reference number | Author |
|---|---|---|
| National Cancer Institute | | Mauricio Comas-Garcia<br>Siddhartha AK Datta<br>Laura Baker<br>Alan Rein |
| Intramural AIDS Targeted Antiviral Therapy Program | | Mauricio Comas-Garcia<br>Alan Rein |
| National Institute of Allergy and Infectious Diseases | | Rajat Varma |
| National Institutes of Health | Contract HHS N26120080001E | Prabhakar R Gudla |

The funders had no role in study design, data collection and interpretation, or the decision to submit the work for publication.

Author contributions
MC-G, Conceptualization, Resources, Data curation, Investigation, Methodology, Writing—original draft, Writing—review and editing; SAKD, Conceptualization, Resources, Writing—review and editing; LB, RV, PRG, Resources, Methodology; AR, Conceptualization, Funding acquisition, Methodology, Writing—original draft, Writing—review and editing

Author ORCIDs
Mauricio Comas-Garcia, http://orcid.org/0000-0002-7733-5138
Alan Rein, http://orcid.org/0000-0002-8273-546X

## Additional files

### Supplementary files
• Supplementary file 1. Diffusion coefficients of the dye used to calibrate the FCS setup and the D of Lysozyme labeled with AF647

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
