## [Decision Letter]

Thank you for submitting your article "Dissection of specific binding of HIV-1 Gag to the "packaging signal" in viral RNA" for consideration by *eLife*. Your article has been favorably evaluated by Gisela Storz (Senior Editor) and three reviewers, one of whom is a member of our Board of Reviewing Editors. The following individual involved in review of your submission has agreed to reveal their identity: Jeremy Luban (Reviewer #3).

The reviewers have discussed the reviews with one another and the Reviewing Editor has drafted this decision to help you prepare a revised submission. In addition to the essential revisions noted below, it will be important for the Board and reviewers to understand the importance of the new conclusions you wish to draw.

Summary:

This paper reports a variety of experiments that score Gag-RNA interactions, as measured by a fluorescence quenching assay, that probe the selectivity of Gag for the viral gRNA, as occurs in assembling virions. This area has a long history, but there remain unanswered questions about this selectivity. The work is carefully done and clearly explained. There are few dramatic new findings: most of the results are confirming earlier work, albeit with better readouts. There are new results concerning Gag-Gag interactions and effects of Gag mutations. Studies of the RNA are not included, except to test monomer and dimer RNAs.

The major gorilla-in-the-room problem is the high background of nonspecific binding manifest in Gag: in many simple in vitro binding experiments, no high selectivity is observed. Here we see that binding at physiological salts again shows no selectivity. Adding competitor tRNA reveals selectivity for short RNAs containing the packaging sequences, (as per earlier work). Mutations that reduce Gag-RNA interactions (MA mutations) help in revealing some selectivity. We now also learn that mutations in Gag that reduce Gag-Gag interactions (CA mutations) reveal some small selectivity, which is enhanced by addition of competitor. The nonspecific binding, and not the specific binding, seems to involve Gag-Gag multimers: Gag mutants with reduced Gag-Gag binding revealed a modest reduction in nonspecific binding (though only 2.5x), and no change in Gag-gRNA binding. High salt also reveals selectivity, phenocopying the CA mutations. The relevant salt concentrations tread a narrow window ledge of enhancing selective binding before all binding is lost.

The assays (by detecting changes in diffusion coefficient) show a new phenomenon going on in Gag binding: the condensation or "collapse" of the RNA. The Gag binding to nonspecific RNAs seems to induce this more effectively. It is unclear exactly what this means, and in particular how many Gags per RNA are involved.

All told these are nice data on a long-standing problem. The bottom line: revealing selective binding requires careful control of the conditions and settings. There are some new conclusions, if no groundbreaking upheavals of dogma.

Essential revisions:

A number of clarifications, listed below, are in order:

1) Reagent validation: Although the lab has much experience in the purification of RNAs and Gag proteins, supplementary figures introduced at the beginning of the Results section to validate the reagents would be desirable. For RNAs, native gels would be useful. For proteins, gels of WT and mutant Gag proteins also would be useful. I'm less concerned about protein purity than about how much protein has aggregated or oligomerized irreversibly at the storage concentration of 20-40 uM.

2) MLV Psi analysis: As a dimeric RNA control, did the authors test a MLV Psi2 dimer ligand in their studies? If so, what were the results?

3) Fraction RNA bound calculation: As far as I can tell from Equation 8, the authors are assuming a linear proportionality of Gag binding to fluorophore quenching, i.e., that binding multiple Gag proteins to an RNA does not have a disproportionate quenching effect relative to binding a single Gag protein. If I am incorrect, the authors might add an explanatory sentence in the data analysis section. If the authors are assuming a linear proportionality, it would be helpful to know why this assumption is correct.

4) Figure 1: Relative to the HIV Psi1, MLV Psi, and GRPE ligands, the fitting of the cooperative binding curve to the diffusion data for the HIV Psi2 ligand appears to be poor. Why?

5) Table 1: The authors have tended to ignore observed fluctuations in the Hill coefficients (nH) and, for the most part, seem to equate every nH of over 1.7 as being roughly equivalent. Is this the case? If not, the authors should explain why tRNA addition decreases the nH of WT Gag binding to HIV Psi1, but increases the nH of WM Gag binding to the same ligand. Other nH variations on the table also should be addressed.

6) tRNA effect on D: The fact that tRNA addition marginally affected WT Gag binding to HIV Psi2, but also reduced "RNA collapse" is consistent with the binding of tRNAs to Gag-HIV Psi2 complexes (subsection “tRNAs reduce the degree of Gag-induced RNA collapse”), but a model for what is happening ought to be included in the Discussion.

7) Figure 9—figure supplement 1: Why is the Kd of the 8N Gag mutant for HIV Psi2 at 500 mM so much better than the Kd of WT Gag?

8) Defining the 310, SSHC, and 8N/WM mutants: This should be done in the first paragraph of the subsection “Decreasing NC-RNA or Gag-Gag interactions reveals specific binding at slightly higher than physiological salt concentrations”.

9) WT Gag-HIV Psi2 binding complexes: Have the authors performed sedimentation or other experiments to determine the Gag:RNA stoichiometries?

---

## [Author Response]

*Essential revisions:*

*A number of clarifications, listed below, are in order:*

*1) Reagent validation: Although the lab has much experience in the purification of RNAs and Gag proteins, supplementary figures introduced at the beginning of the Results section to validate the reagents would be desirable. For RNAs, native gels would be useful. For proteins, gels of WT and mutant Gag proteins also would be useful. I'm less concerned about protein purity than about how much protein has aggregated or oligomerized irreversibly at the storage concentration of 20-40 uM.*

Three new supplementary figures, linked to Figure 1, have been added. Figure 1—figure supplement 1 shows two characteristic denaturing gels for the purified labeled RNAs (6% polyacrylamide/TBE-UREA gel). These gels show that the in vitro transcripts are intact, without significant levels of early-termination products or degradation products. Figure 1—figure supplement 1 shows a native gel as suggested by the reviewer, showing that the RNA is not aggregating significantly, even after being incubated for dimerization.

Figure 1—figure supplement 2 shows data from analytical ultracentrifugation of Gag at 1 μM. This protein was stored at 20 μM, thawed and diluted to 1 μM. This data shows that 97% of the absorbance corresponds to a sedimentation coefficient expected for monomeric Gag. Furthermore, it shows that there are no detectable aggregates due to freezing/thawing cycles.

Figure 1—figure supplement 2 shows the normalized autocorrelation function of Alexa Fluor 488-labeled HIV-1 WT Δp6 Gag (at 300 nM). As in the case of the RNAs the autocorrelation function shows that at this protein concentration there are no detectable aggregates that could have been produced during freezing. The diffusion coefficient for Gag measured by FCS is ≈ 106 μm^[13]^/s, thus by using the Stokes-Einstein equation (which assumes a spherical volume) we obtained a hydrodynamic radius (***R_h_***) ≈ 2.5 nm. This shows that at under these experimental conditions Gag is monomeric and that there is no significant aggregation in solution. Figure 1—figure supplement 2 shows a 4-12% NuPAPGE SDS-PAGE gel for WT and all the mutant Gag proteins.

Further, Figure 1—figure supplement 3 shows the normalized autocorrelation functions for the monomeric and dimeric HIV-1 Ψ, GRPE and MoMLV Ψ (panels A, B, C and D, respectively). The fact that the value of autocorrelation functions at lag times where large aggregates are to be expected (Lag times between 0.01 and 0.1 s) is zero indicates that at 15 nM these RNAs do not form irreversible aggregates that could have been caused by freezing and thawing cycles or by the RNA dimerization protocol.

Finally, this information has been added to the Materials and methods section (subsection “RNA in vitro transcription and RNA labeling”, subsection “Protein expression and purification”, and subsection “Fluorescence correlation spectroscopy (FCS) and data analysis”).

*2) MLV Psi analysis: As a dimeric RNA control, did the authors test a MLV Psi2 dimer ligand in their studies? If so, what were the results?*

We did not test dimeric MoMLV Ψ in our studies. The main reason for this was that even for HIV-1 Ψ the difference in the binding character between monomeric and dimeric RNAs was small. In fact, the only condition where the monomeric HIV-1 Ψ showed a significantly impaired binding character was in the tRNA competition experiments using WM Gag. This has been addressed in the fifteenth paragraph of the Discussion.

*3) Fraction RNA bound calculation: As far as I can tell from Equation 8, the authors are assuming a linear proportionality of Gag binding to fluorophore quenching, i.e., that binding multiple Gag proteins to an RNA does not have a disproportionate quenching effect relative to binding a single Gag protein. If I am incorrect, the authors might add an explanatory sentence in the data analysis section. If the authors are assuming a linear proportionality, it would be helpful to know why this assumption is correct.*

This is a crucial question. Our answer has two parts. First, because of the nature of the interaction of protein with the fluorophore, the quenching is all-or-none. That is, as mentioned in the manuscript (Results, third paragraph), excitation of Cy5 induces a reversible conversion to a completely non-fluorescent state, evidently stabilized by steric interactions in the Gag-RNA complexes. This is distinct from the quenching experienced by many fluorophores, in which there is a change in emission spectra or the relaxation time of the dye (see Levitus and Ranjit, 2011 and Stennett, Ciuba and Levitus 2014). Since the conversion is all-or-none, there is a direct, linear relationship between the number of molecules quenched and the decrease in total fluorescence.

The second part of the answer is to point out that the quenching eventually reaches a plateau, where further addition of Gag protein has no further effect on fluorescence; we interpret this to mean that at this plateau, all the RNA molecules are bound by one or more Gag molecules. This stoichiometry is taken as 100% bound RNA in the Gag titrations. Although these are the only assumptions we used in converting the quenching data into binding plots, it should be noted that a model positing cooperative binding gave a far better fit to the binding data than other models (Figure 1—figure supplement 6).

We have discussed these aspects of the data analysis in the Supplementary Information subsection “Fluorescence spectroscopy and data analysis”.

*4) Figure 1: Relative to the HIV Psi1, MLV Psi, and GRPE ligands, the fitting of the cooperative binding curve to the diffusion data for the HIV Psi2 ligand appears to be poor. Why?*

This is probably because the number of data points in the sigmoidal portion of the curve is very small. This increases the error of the fit. This issue has been addressed in the last paragraph of the subsection “Gag binding causes RNA collapse”.

*5) Table 1: The authors have tended to ignore observed fluctuations in the Hill coefficients (nH) and, for the most part, seem to equate every nH of over 1.7 as being roughly equivalent. Is this the case? If not, the authors should explain why tRNA addition decreases the nH of WT Gag binding to HIV Psi1, but increases the nH of WM Gag binding to the same ligand. Other nH variations on the table also should be addressed.*

The reviewers are correct that we have not distinguished between the different n_H_ values we obtained. We acknowledge the imprecision in the measurements and note this in the last paragraph of the subsection “Gag binds to RNAs with high affinity but low specificity”.

*6) tRNA effect on D: The fact that tRNA addition marginally affected WT Gag binding to HIV Psi2, but also reduced "RNA collapse" is consistent with the binding of tRNAs to Gag-HIV Psi2 complexes (subsection “tRNAs reduce the degree of Gag-induced RNA collapse”), but a model for what is happening ought to be included in the Discussion.*

We have added the following model to the Discussion: “It was interesting to note that the presence of excess yeast tRNA greatly reduced the magnitude of RNA collapse (Figure 3—figure supplement 1). This is even true with 8N Gag (Figure 5), for which binding per seis almost unaffected by tRNA addition (Figure 5). These observations suggest that tRNA is incorporated into the complexes of Gag and tagged RNA, reducing compaction, viaone or more Gag domains other than MA.”

*7) Figure 9—figure supplement 1: Why is the Kd of the 8N Gag mutant for HIV Psi2 at 500 mM so much better than the Kd of WT Gag?*

We have added the following explanation to the Discussion section: “It is somewhat surprising that the apparent affinity of 8N Gag for the dimeric HIV-1 Ψ between 500 and 550 mM NaCl is greater than that of WT Gag (Figure 9—figure supplement 3). However, the interaction of the WT MA domain with RNAs is presumably almost exclusively electrostatic and is thus negligible at these high salt concentrations; we surmise that under these conditions, repulsion between positively charged WT MA domains of two or more RNA-bound Gag molecules raises the energetic cost of RNA-binding, and that this repulsion is reduced by the 8N mutations.”

8) Defining the 310, SSHC, and 8N/WM mutants: This should be done in the first paragraph of the subsection “Decreasing NC-RNA or Gag-Gag interactions reveals specific binding at slightly higher than physiological salt concentrations”.

We have now introduced these mutants in the first paragraph of the subsection “Decreasing NC-RNA or Gag-Gag interactions reveals specific binding at slightly higher than physiological salt concentrations”.

*9) WT Gag-HIV Psi2 binding complexes: Have the authors performed sedimentation or other experiments to determine the Gag:RNA stoichiometries?*

We have not performed such experiments. The problem is that sedimentation experiments, as well as most biophysical techniques, require higher Gag concentrations; at these concentrations the Gag/RNA complexes will associate into virus-like-particles. We have addressed this at the beginning of the Results section.